# Confidence Regulation Neurons in Language Models

**Alessandro Stolfo**[*]
ETH Zürich

**Ben Wu**[*]
University of Sheffield

**Wes Gurnee**
MIT

**Yonatan Belinkov**
Technion

**Xingyi Song**
University of Sheffield

**Mrinmaya Sachan**
ETH Zürich

**Neel Nanda**

## Abstract

Despite their widespread use, the mechanisms by which large language models (LLMs) represent and regulate uncertainty in next-token predictions remain largely unexplored. This study investigates two critical components believed to influence this uncertainty: the recently discovered entropy neurons and a new set of components that we term token frequency neurons. Entropy neurons are characterized by an unusually high weight norm and influence the final layer normalization (LayerNorm) scale to effectively scale down the logits. Our work shows that entropy neurons operate by writing onto an *unembedding null space*, allowing them to impact the residual stream norm with minimal direct effect on the logits themselves. We observe the presence of entropy neurons across a range of models, up to 7 billion parameters. On the other hand, token frequency neurons, which we discover and describe here for the first time, boost or suppress each token's logit proportionally to its log frequency, thereby shifting the output distribution towards or away from the unigram distribution. Finally, we present a detailed case study where entropy neurons actively manage confidence in the setting of induction, i.e. detecting and continuing repeated subsequences.

## 1 Introduction

As large language models (LLMs) increasingly permeate high-stakes applications, the lack of transparency in their decision-making processes poses significant vulnerabilities and risks [5]. Understanding the basis of these models' decisions, especially how they regulate confidence in their predictions, is crucial not only for advancing model development but also for ensuring their safe deployment [1, 33]. LLMs have been empirically shown to be fairly well calibrated: on question-answering tasks, token-level probabilities of a model prediction generally match the probability of the model being correct [40, 58]. This raises the question of whether LLMs possess mechanisms for general-purpose calibration to mitigate the risks associated with overconfident predictions.

Significant research has been conducted on estimating model uncertainty in neural networks [19, 21] and specifically in LLMs [22]. While many studies focus on quantifying and calibrating model confidence [49, 40, 47, 45, 65, *inter alia*], there is little research into the internal mechanisms LLMs might use to calibrate their predictions.

In this work, we explore two types of components in Transformer-based language models that we believe serve a calibration function: the recently identified entropy neurons and a new class of components that we term *token frequency neurons*. Entropy neurons have been brought to light by recent work [42, 28] and are characterized by their high weight norm and their low composition with the unembedding matrix despite being in the final layer. The low composition with the unembedding matrix would suggest that they play a minor role in the next-token prediction.

---

[*]Equal contribution. Correspondence to `stolfoa@ethz.ch`, `bpwu1@sheffield.ac.uk`.

38th Conference on Neural Information Processing Systems (NeurIPS 2024).

However, their high weight norm, despite LLMs being trained with weight decay [48], indicates that these neurons must be important for performance. This leads us to infer that these neurons may play a calibration role, as hypothesized by Gurnee et al. [28]. We show that entropy neurons work by writing to an effective null space of the unembedding matrix and leverage the layer normalization [4] that is applied to the residual stream. This way, these components effectively modulate the entropy of the model's output distribution in a manner that only minimally affects the actual model prediction (Figure 1). We observe the presence of entropy neurons in GPT-2 [62], Pythia [6], Phi-2 [38], Gemma 2B [68], and LLaMA2 7B [70], demonstrating for the first time their role across different model families and scales.[2]

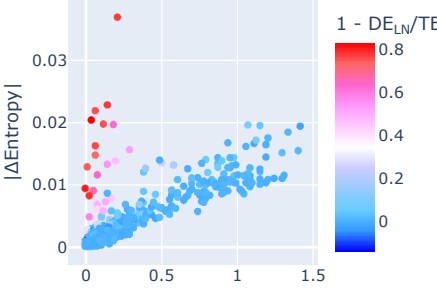

Figure 1: **Entropy and Prediction.** We mean ablate final-layer neurons across 4000 tokens and measure the variation in the entropy of the model's output $P_{\text{model}}$ against average change of model's prediction ($\text{argmax}_x P_{\text{model}}(x)$). We identify a set of neurons whose effect depends on LayerNorm (red points; metric described in §3.2), and which affect the model's confidence (quantified as entropy of $P_{\text{model}}$) with minimal impact on the prediction.

Motivated by the importance of the token frequency distribution (i.e., the unigram distribution) in language modeling [11, 51], we identify a novel class of neurons that boost or suppress each token's logit proportionally to its frequency. We provide evidence showing that these neurons, which we term token frequency neurons, modulate the distance between the model's output distribution and the unigram distribution, which the model defaults to in settings of high uncertainty.

As a case study, we analyze the activation of these neurons in scenarios involving induction (i.e., the repetition of subsequences in the input; 57). Our results show that entropy neurons increase the output distribution's entropy, thus decreasing the model's confidence in its predictions for repeated sequences. This represents a hedging mechanism, aimed at mitigating loss spikes on confidently wrong predictions.

Our contributions are as follows: (1) We show that models have dedicated circuitry to calibrate their confidence, with dedicated neurons in the final layer: the previously discovered entropy neurons and novel token frequency neurons. (2) We explore the mechanism by which each neuron family affects confidence (§3 and §4). Notably, we find that the model learns a low-rank effective null space for the unembedding that entropy neurons write to, and the final LayerNorm scale is used to calibrate the output logits. This striking phenomenon would have been missed by the assumptions made in standard methods such as direct logit attribution [16, 24, 46]. (3) We study how these neurons are used in practice to regulate confidence (§5). As expected, neurons that lower confidence worsen performance when models are correct and improve it when models are incorrect. We close with a mechanistic case study of how induction heads [57] use entropy neurons to control confidence when detecting and continuing repeated text (§6).[3]

## 2 Background

In this section, we provide an overview of the Transformer architecture [71], focusing on the components relevant to our analysis. Given a vocabulary $\mathcal{V}$, we denote an autoregressive language model as $\mathcal{M} : \mathcal{X} \rightarrow \mathcal{Y}$, where $\mathcal{Y}$ is the space of probability distributions over $\mathcal{V}$. $\mathcal{M}$ takes as input a token sequence $x = [x_1, ..., x_m] \in \mathcal{X} \subseteq \mathcal{V}^m$, and outputs a probability distribution $P_{\text{model}} : \mathcal{V} \rightarrow [0, 1]$ to predict the next token in the sequence. In this work, we focus on decoder-only Transformer-based models, which represent the backbone of the most capable current AI systems [58, 26]. The Transformer architecture consists of two core components: the multi-head self-attention and a multi-layer perceptron (MLP). These two components read information in from and write out to the residual stream (i.e., the per-token hidden state consisting of the sum of all previous component outputs) [16].

---

[2]In the main paper, we report detailed results for GPT-2 Small and LLaMA2 7B. Results for other models are provided in the Appendix.

[3]Our code and data are available at `https://github.com/bpwu1/confidence-regulation-neurons`.

**MLP.** Central to this study is the structure of the MLP layers of the transformer.[4] Given a normalized residual stream hidden state $\mathbf{x} \in \mathbb{R}^{d_{\text{model}}}$, the output of an MLP layer is

$$\text{MLP}(\mathbf{x}) = \mathbf{W}_{\text{out}}\sigma(\mathbf{W}_{\text{in}}\mathbf{x} + \boldsymbol{\beta}_{\text{in}}) + \boldsymbol{\beta}_{\text{out}} , \tag{1}$$

where $\mathbf{W}_{\text{out}}^T, \mathbf{W}_{\text{in}} \in \mathbb{R}^{d_{\text{mlp}} \times d_{\text{model}}}$ are learned weight matrices, $\boldsymbol{\beta}_{\text{in}}$ and $\boldsymbol{\beta}_{\text{out}}$ are learned biases. The function $\sigma$ denotes an element-wise nonlinear activation function, typically a GeLU [32]. In this paper, the term *neuron* refers to an entry in the MLP hidden state. We denote specific neurons in the model using the format `<layer>.<index>`. We use $\mathbf{n}$ to refer to the *activation values* of the neurons (i.e., the output of the activation function $\sigma$). Furthermore, we use $\mathbf{w}_{\text{out}}^{(i)} \in \mathbb{R}^{d_{\text{model}}}$ to indicate the output weights of neuron $i$ (i.e., the $i$-th column of $\mathbf{W}_{\text{out}}$).

**LayerNorm.** Layer normalization (LayerNorm) is a commonly employed technique to improve the stability of the training process in deep neural networks [4]. Given a hidden state $\mathbf{x} \in \mathbb{R}^{d_{\text{model}}}$, LayerNorm applies the transformation

$$\text{LN}(\mathbf{x}) = \frac{\mathbf{x} - m(\mathbf{x})}{\sqrt{\text{Var}(\mathbf{x}) + \epsilon}} \odot \boldsymbol{\gamma} + \boldsymbol{\beta}. \tag{2}$$

In words, LayerNorm centers a hidden state by subtracting its mean $\left(m(\mathbf{x}) = \frac{1}{d_{\text{model}}}\sum_j \mathbf{x}_j\right)$, re-scales it by its $\epsilon$-adjusted standard deviation ($\sqrt{\text{Var}(\mathbf{x}) + \epsilon}$, where $\epsilon \in \mathbb{R}$), and applies an element-wise affine transformation determined by the learned parameters $\boldsymbol{\gamma}, \boldsymbol{\beta} \in \mathbb{R}^{d_{\text{model}}}$.

**Unembedding.** After the final transformer block, the final state of the residual stream is passed through a final LayerNorm and then projected onto the vocabulary space through an unembedding matrix $\mathbf{W}_{\text{U}} \in \mathbb{R}^{|\mathcal{V}| \times d_{\text{model}}}$. The logits obtained by mapping the residual stream final state onto the vocabulary space are then passed through a softmax function to convert them into a probability distribution $P_{\text{model}} \in \mathcal{Y}$. Finally, the loss is computed using a function $\mathcal{L} : \mathcal{Y} \times \mathcal{V} \to \mathbb{R}_+$ (typically cross entropy for autoregressive language models). Following previous work [16], we employ standard weight pre-processing techniques [53]. In particular, we "fold" LayerNorm trainable parameters into the $\mathbf{W}_{\text{in}}$ and $\mathbf{W}_{\text{U}}$ matrices. We report the details about weight pre-processing in Appendix C.

## 3 Entropy neurons

Prior research examining individual neurons in GPT-2 language models [62] identified a category of neurons notable for their high norm and limited interaction with the unembedding matrix. Katz and Belinkov [42] speculate that these neurons may act as regularizers, contributing a constant bias to the residual stream. Gurnee et al. [28] independently rediscovered these neurons, termed them *entropy neurons*, and suggest that they play a role in regulating the entropy of the model's output distribution. They show that interventions increasing the activation value of these neurons affect the entropy of the output distribution to a larger extent compared to other neurons. Despite these insights, the natural activation effects of these neurons on standard language inputs have not been examined, and *how* they carry out this entropy-modulating function remains unclear. In this section, we explore the mechanism of action of these neurons within the model.

### 3.1 Identifying entropy neurons

Our initial step is to identify which neurons are entropy neurons. We do this by searching for neurons with a high weight norm and a minimal impact on the logits. To detect minimal effect on the logits, we follow the heuristic used by Gurnee et al. [28], analyzing the variance in the effect of neurons on the logits. In particular, we project the last-layer neuron weights in GPT-2 Small onto the vocabulary space. This projection, sometimes referred to as logit attribution, represents an approximation of the neuron's effect on the final prediction logits [55, 24, 15]. Then, we compute the variance of the normalized projection. That is, given a neuron with output weights $\mathbf{w}_{\text{out}}$, we compute

$$\text{LogitVar}(\mathbf{w}_{\text{out}}) = \text{Var}\left(\frac{\mathbf{W}_{\text{U}}\mathbf{w}_{\text{out}}}{\|\mathbf{W}_{\text{U}}\|_{\text{dim}=1}\|\mathbf{w}_{\text{out}}\|}\right), \tag{3}$$

---

[4]In particular, our study focuses on the final-layer MLPs due to their direct path to the final LayerNorm and unembedding, which excludes possible interactions with other transformer layers.

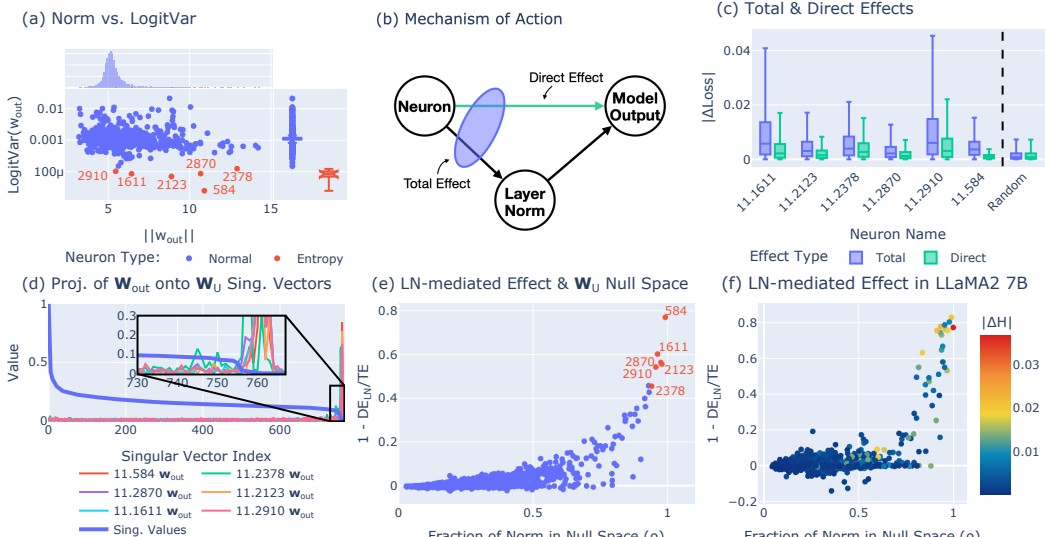

Figure 2: **Identifying and Analyzing Entropy Neurons.** (a) Neurons in GPT-2 Small displayed by their weight norm and variance in logit attribution. Entropy neurons (red) have high norm and low logit variance. (b) Causal graph showing the total effect and direct effect (bypassing LayerNorm) of a neuron on the model's output. (c) Comparison of total and direct effects on model loss for entropy neurons and randomly selected neurons. (d) Singular values and cosine similarity between neuron output weights and singular vectors of $\mathbf{W}_U$. (e) Entropy neurons (red) show significant LayerNorm-mediated effects and high projection onto the null space ($\rho$). (f) Relationship between $\rho$ and the LayerNorm-mediated effect in LLaMA2 7B. $\rho$ is computed with $k = 40 \approx 0.01 * d_{\text{model}}$. Color represents absolute change in entropy upon ablation ($\Delta$H).

where $\| \cdot \|_{\text{dim}=1}$ indicates column-wise norm.

This measure allows us to quantify how specific is the direct effect of a neuron on the output logits, with the underlying intuition that a diffused contribution (i.e., close to a constant added to all logits before applying the softmax) results in a small effect on the output probabilities. In the last layer of GPT-2 Small, we identify a subset of neurons with particularly low variance but substantial norm (Figure 2a). Since the optimization of the model's weights preserves these high-norm low-composition neurons (that are expensive in terms of weight decay penalty [48]), it is likely that these neurons are important and exert their effect through a different mechanism than directly modifying the logits.

## 3.2 Mechanism of action

Given the low composition with the unembedding matrix, we hypothesize that most of the effect of entropy neurons on the model's output is mediated by the re-scaling of the residual stream performed by the final LayerNorm. We test this hypothesis using a causal mediation analysis formulation [60, 64], illustrated in Figure 2b. We distinguish the *total* effect that a neuron has on the model's output (represented by the two causal paths stemming from the neuron node in Figure 2b) from its *direct* effect (the green arrow in Figure 2b), which is not mediated by the change in the LayerNorm scale. Our hypothesis posits that the difference between the former and the latter is significantly larger for entropy neurons than for normal neurons.

We carry out an ablation experiment where we intervene on the activation value of a specific neuron by fixing it to its mean value across a reference distribution while constraining the LayerNorm scaling coefficient to remain constant. More formally, consider a neuron with index $i \in \{1, 2, \ldots, d_{\text{mlp}}\}$ and denote by $\mathbf{n}_i \in \mathbb{R}$ its activation value. Given an input $x \in \mathcal{X}$, denote by $\mathbf{x}$ the last hidden state in the model (i.e., the output of the last transformer block), and denote by $\mathbf{x}^{-i}$ the last hidden state in the model obtained after mean-ablating neuron $i$:

$$\mathbf{x}^{-i} = \mathbf{x} + (\overline{\mathbf{n}}_i - \mathbf{n}_i)\mathbf{w}_{\text{out}}^{(i)}, \tag{4}$$

where $\overline{\mathbf{n}}_i$ is the mean activation value computed over a subset of $\mathcal{X}$.

We quantify the total effect of neuron $i$ upon mean ablation by measuring the absolute variation in the model's loss $\mathcal{L}$,[5] specifically:

$$\text{TE}(i) = \mathbb{E}_x\left[\left|\mathcal{L}\left(\mathbf{W}_\text{U}\text{LN}(\mathbf{x}), x\right) - \mathcal{L}\left(\mathbf{W}_\text{U}\text{LN}(\mathbf{x}^{-i}), x\right)\right|\right], \tag{5}$$

where the expectation is taken over a uniformly sampled subset of a corpus $\mathcal{X}$. Similarly, we quantify the direct effect of neuron $i$ by preventing the LayerNorm denominator from varying:

$$\text{DE}_\text{LN}(i) = \mathbb{E}_x\left[\left|\mathcal{L}\left(\mathbf{W}_\text{U}\text{LN}(\mathbf{x}), x\right) - \mathcal{L}\left(\mathbf{W}_\text{U}\left(\frac{\mathbf{x}^{-i} - m(\mathbf{x}^{-i})}{\sqrt{\text{Var}(\mathbf{x}) + \epsilon}} \odot \boldsymbol{\gamma} + \boldsymbol{\beta}\right), x\right)\right|\right], \tag{6}$$

For each neuron, we measure the total and direct effects when its activation value is set to the mean across a dataset of 25600 tokens from the C4 Corpus [63] (additional experimental details are provided in Appendix D). We compare these metrics for six selected entropy neurons against those from 100 randomly selected neurons. The results (Figure 2c) show that in the randomly selected neurons there is no significant difference between their total and direct effects. On the other hand, the difference between these two quantities is substantial for entropy neurons, and in some cases the direct effect represents only a small fraction of the total effect.

These findings represent an interesting and novel example of how language models can use LayerNorm to indirectly manipulate the logit values. Such a mechanism could easily be overlooked by conventional analyses, which typically focus on direct logit attribution and fail to account for the normalization effects imposed by LayerNorm [16, 24, 46].

### 3.3 The unembedding has an effective null space

The unembedding is a linear map to a significantly higher-dimensional space (e.g., from 768 to 50,257 dimensions in GPT-2). Therefore, it is surprising that the output of a high-norm neuron would have little effect on the logits. We hypothesize that there is a subspace of the residual stream with an unusually low impact on the output that entropy neurons write onto. To investigate this, we compute the singular value decomposition (SVD) of the unembedding matrix $\mathbf{W}_\text{U} = \mathbf{U}\boldsymbol{\Sigma}\mathbf{V}^T$.

Analyzing the singular values in $\boldsymbol{\Sigma}$ (thick blue line in Figure 2d), we observe that the bottom values are extremely small. In particular, we notice a sharp drop to values close to 0 around index 755. This observation indicates a remarkable phenomenon within the model: the training procedure optimizes the weights to preserve what effectively functions as a null space within $\mathbf{W}_\text{U}$. This finding is striking, as it suggests that the model deliberately limits its representational power by projecting a set of residual stream directions onto a small neighborhood of the 0-vector.

Next, we study the directions within the residual stream that entropy neurons write to. We do this by computing the cosine similarity between the output weights ($\mathbf{w}_\text{out}$) for each neuron and singular vectors $\mathbf{V} \in \mathbb{R}^{d_\text{model} \times d_\text{model}}$. The results, illustrated by the colored lines in Figure 2d, show that all the entropy neurons write almost exclusively to directions within the effective null space. These findings illustrate how entropy neurons leverage this null space to add norm to the residual stream without directly affecting the logit values.

### 3.4 Universality across model families

The relationship observed between entropy neurons and the unembedding null space suggests that entropy neurons may be identified by analyzing the proportion of their norm that is projected onto such a null space. To test this, given a neuron $i$, we compute the fraction $\rho_i$ of the neuron's norm projected onto the effective null space $\mathbf{V}_0^T$. In particular, we define the effective null space of $\mathbf{W}_\text{U}$ as the space spanned by its bottom $k$ singular vectors, i.e., $\mathbf{V}_0^T := \langle \mathbf{V}_{:,d_\text{model}-k}^T, \ldots, \mathbf{V}_{:,d_\text{model}}^T \rangle$, and we compute $\rho_i = \|\mathbf{V}_0^T\mathbf{w}_\text{out}^{(i)}\|/\|\mathbf{w}_\text{out}^{(i)}\|$. Then, we measure the quantity $1 - \text{DE}(i)/\text{TE}(i)$, which represents the fraction of the neuron's effect that is mediated by the final LayerNorm.[6] We

---

[5]For readability, here we omit the final softmax to normalize the logits before computing the loss.

[6]We opt for representing the LayerNorm-mediated effect by $1 - \text{DE}(i)/\text{TE}(i)$, rather than computing the indirect effect of the LayerNorm scale because a large indirect effect might highlight neurons with high norm *and* large direct effect, such as a neuron that boosts a specific logit. In our setting, we are interested in neurons that actually have low direct effect, which is not implied by a large indirect effect.

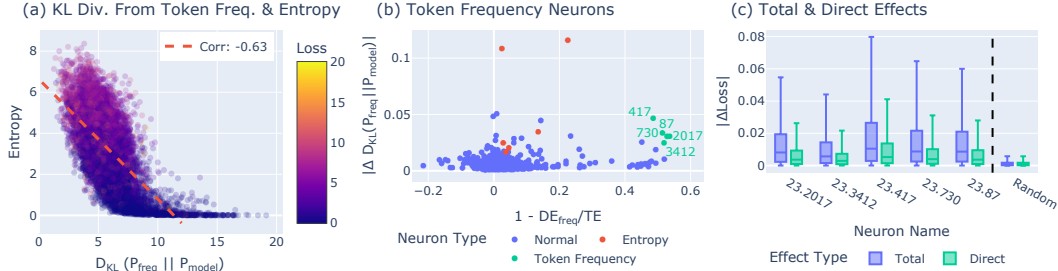

Figure 3: **Token Frequency Neurons in Pythia 410M.** (a) $D_{KL}(P_{freq}\|P_{model})$ and Entropy are correlated negatively. (b) Scatter plot of neurons highlighting token frequency neurons (in green), with high effect on $D_{KL}(P_{freq}\|P_{model})$, significantly mediated by the token frequency direction. (c) Box plots showing substantial difference in total vs. direct effect in token frequency neurons.

compare these two quantities for each neuron at the last layer in GPT-2 Small (Figure 2e), with $k = 12$. The results show that neurons with the largest projections onto the null space have the most significant LayerNorm-mediated effects on the model's output. These findings connect with recent work highlighting the importance of the bottom $\mathbf{W}_U$ singular vectors [10] and represent further evidence supporting our hypothesis about the entropy neurons' mechanism of action.

We extend our investigation of entropy neurons to models beyond the GPT-2 family. In Figure 2f, we analyze the proportion of norm projected onto the bottom singular vectors of $\mathbf{W}_U$ (i.e., $\rho$) and the influence of LayerNorm on the output for neurons in the last layer of LLaMA2 7B [70].[7] Similar to our observations with GPT-2 Small, the results indicate the existence of a distinct group of neurons that predominantly write onto the effective null space of $\mathbf{W}_U$. These neurons significantly influence the model's output via LayerNorm, confirming the presence of entropy neurons across different model families. As entropy neurons act by re-scaling the residual stream and have no effect that is specific to a subset of the tokens, their activation has a large effect on the entropy of the output distribution but little impact on the relative ranking of the tokens. That is, they affect the model's confidence in the prediction without affecting the actual token being predicted. This is illustrated in Figure 1. We repeat these analyses on GPT-2 Medium, Pythia 410M and 1B[6], Phi 2 [38], and Gemma 2B [68], obtaining overall consistent results (Appendix F).

## 4 Token frequency neurons

The token frequency distribution (i.e., the distribution of unigrams over a corpus) can serve as a reliable baseline for next-token prediction, especially in scenarios of high uncertainty [11, 51]. Thus, we hypothesize that a useful general-purpose mechanism for regulating confidence in language models could involve modulating the distance between the output distribution and the token frequency distribution. We explore this hypothesis using the 410M-parameter Pythia model, for which the training corpus has been publicly released (The Pile; 20). For this corpus, we compute the empirical token frequency distribution $P_{freq}$. We observe that the entropy of the model's output distribution is in fact negatively correlated with its Kullback-Leibler (KL) divergence from the empirical token frequency distribution: as the confidence of the model decreases (higher entropy), its output distribution tends to be closer to the empirical token frequency distribution (Figure 3a).

As for entropy neurons, we aim to identify neurons whose operational mechanism relies on a particular subspace—in this case, the direction corresponding to the token frequency distribution. We obtain such a direction by computing the logit vector $\mathbf{v}_{freq} \in \mathbb{R}^{|\mathcal{V}|}$ by centering the log-probability values for each token in $\mathcal{V}$. That is, $\mathbf{v}_{freq,i} = \log(\mathbf{p}_{freq,i}) - m(\log(\mathbf{p}_{freq}))$, where $\mathbf{p}_{freq} \in \mathbb{R}^{|\mathcal{V}|}$ the vector of the token frequency probabilities values, and $m : \mathbb{R}^{|\mathcal{V}|} \to \mathbb{R}$ indicates the mean function.

To identify the neurons that rely on this direction to modulate the model's output, we conduct an ablation experiment similar to the one described in §3.2. In this experiment, we mean-ablate neurons and assess the absolute change in model loss while preserving the value of the logits along the

---

[7]LLaMA actually uses RMSNorm [80] which differs from LayerNorm in the absence of re-centering and bias term. However, our experimental procedure is not affected by this difference.

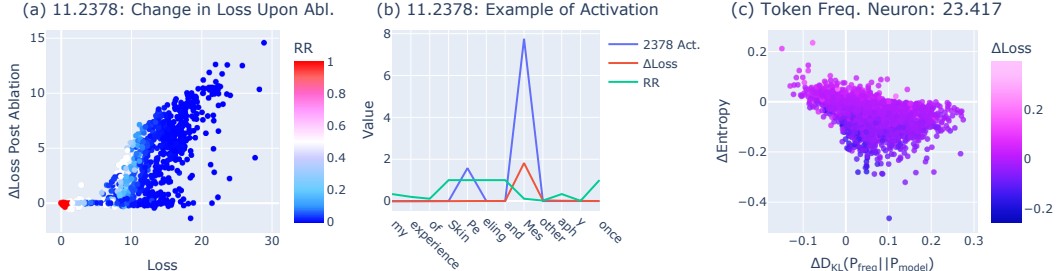

Figure 4: **Examples of Neuron Activity in Language Models.** (a) Change in loss after ablation of entropy neuron 11.2378 in GPT-2 Small. Color indicates reciprocal rank ($\mathrm{RR}$) of the correct token prediction. (b) Activation of neuron 11.2378 on an example from the C4 Corpus. The neuron mitigates a loss spike at the token "Mes," after which the model predicts "otherapy." (c) Change in entropy and KL divergence on correct tokens ($\mathrm{RR} = 1$) post ablation of neuron 23.417 in Pythia 410M. The neuron increases entropy and aligns the model's output with the token frequency distribution.

direction of $\mathbf{v}_{\text{freq}}$ as constant. Denote by $\mathbf{l} = \mathbf{W}_{\text{U}}\text{LN}(\mathbf{x})$ the logits produced by the model on input $x \in \mathcal{X}$, and denote by $\mathbf{l}^{-i} = \mathbf{W}_{\text{U}}\text{LN}(\mathbf{x}^{-i})$ the value of the logits obtained on the same input after the ablation of a neuron $i$ (where $\mathbf{x}^{-i}$ is defined in Eq. (4)). To measure the effect of neuron $i$ that is not mediated by the token frequency direction (i.e., its *direct* effect), we obtain the adjusted value of the post-ablation logits $\mathbf{l}^{-i}$ as

$$\tilde{\mathbf{l}}^{-i} = \mathbf{l}^{-i} + \frac{(\mathbf{l}^{T}\mathbf{v}_{\text{freq}}) - (\mathbf{l}^{-iT}\mathbf{v}_{\text{freq}})}{\|\mathbf{v}_{\text{freq}}\|^{2}}\mathbf{v}_{\text{freq}}. \tag{7}$$

Then we compute the total and direct effects of neuron $i$ as

$$\text{TE}(i) = \mathbb{E}_{x \sim \mathcal{X}}\left[\left|\mathcal{L}\Big(\mathbf{l}, x\Big) - \mathcal{L}\Big(\mathbf{l}^{-i}, x\Big)\right|\right], \text{ and } \text{DE}_{\text{freq}}(i) = \mathbb{E}_{x \sim \mathcal{X}}\left[\left|\mathcal{L}\Big(\mathbf{l}, x\Big) - \mathcal{L}\Big(\tilde{\mathbf{l}}^{-i}, x\Big)\right|\right]. \tag{8}$$

In Figure 3b, we report the results comparing the token frequency-mediated effect against the average absolute change in the KL divergence between $P_{\text{model}}$ and $P_{\text{freq}}$ for neurons at the final layer of Pythia 410M. We also highlight the entropy neurons with the highest LayerNorm-mediated effect (see Appendix F for comprehensive results on entropy neurons in Pythia 410M). We observe that there are neurons whose effect on $P_{\text{model}}$ is substantially mediated by the token frequency direction (i.e., positive value along the $x$-axis in Figure 3b). These neurons, upon ablation, lead to significant variation in $\mathrm{D}_{\text{KL}}(P_{\text{freq}}\|P_{\text{model}})$ (large positive value along the $y$-axis in Figure 3b), comparable to the variations caused by some entropy neurons.

These results validate the presence of components (which we term *token frequency neurons*) that affect the model's output distribution by bringing it closer or away from the token frequency distribution by *directly* modulating it. Figure 3c focuses on 5 selected token frequency neurons and shows their impact on the loss and its decrease upon the inhibition of the token frequency direction (Eq. (7)), which parallels the LayerNorm mediation for entropy neurons discussed in §3. We additionally show the presence of token frequency neurons in GPT-2 Small and Pythia 1B in Appendix G.

## 5 Examples of neuron activity

We have shown that confidence-regulation neurons improve model performance by calibrating its output across the entire distribution. However, it remains unclear what this looks like in practice. To better illustrate the role that these neurons play in language models, we provide additional analyses about the changes in the model output induced by particular entropy and token frequency neurons.

**Entropy neurons.** In GPT-2 Small, we examine the change in entropy induced by one of the strongest entropy neurons identified in §3 (11.2378). To study this, we conduct an experiment in which we clip the activation value of the neuron to its mean and measure the resulting change in the model output.[8] We analyze the difference in loss caused by the ablation of the neuron on

---

[8] "Clipping" refers to setting the activation value of the neuron to its mean only when it exceeds it, unlike mean ablation, which sets the activation value to its mean regardless.

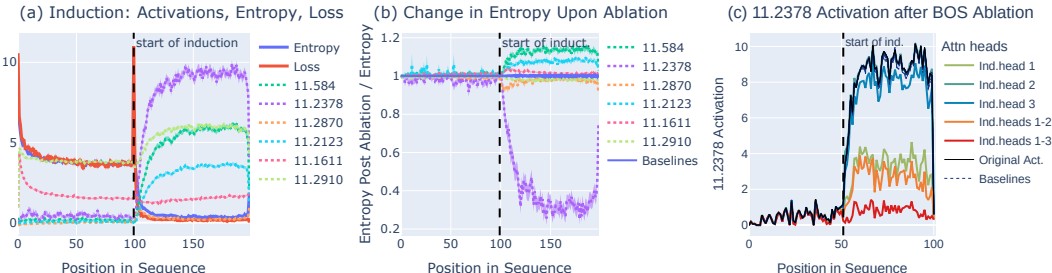

Figure 5: **Entropy Neurons on Induction.** (a) Activations, entropy, and loss across duplicated 200-token input sequences. (b) The effect of clip mean-ablation of specific entropy neurons. Neuron 11.2378 shows the most significant impact, with up to a 70% reduction in entropy. (c) BOS ablation of induction heads: Upon the ablation of three induction heads in GPT-2 Small, the activation of entropy neuron 11.2378 decreases substantially.

individual tokens and compare it to the initial loss value. This analysis is depicted in Figure 4a, which also illustrates the reciprocal rank of the correct token prediction in the model's output distribution (RR). These observations suggest that the entropy-increasing function of neuron 11.2378 acts as a hedging mechanism: it marginally increases the loss on correct model predictions (low initial loss) but prevents the loss from spiking when the model confidently predicts a wrong token (high initial loss). In Figure 4b, we provide an example of input, taken from the C4 corpus [63], on which the model confidently predicts a wrong token (*"otherapy"*), and neuron 11.2378 activates to mitigate the loss spike. We perform the same analysis for the strongest entropy neuron in LLaMA2 7B in Appendix F.1.

**Token frequency neurons.** We next focus on neuron 23.417 in Pythia 410M (identified in §4). This is a token frequency neuron: when its activation is increased, it pushes the model's output $P_{\text{model}}$ towards the token frequency distribution $P_{\text{freq}}$. This is illustrated by Figure 4c, showing the KL divergence between the two distributions pre- and post-ablation. As with increased entropy, bringing $P_{\text{model}}$ closer to $P_{\text{freq}}$ decreases the model's confidence (positive values along the $x$-axis correlate with negative $y$ values in Figure 4c). This leads to an increase in loss on correct predictions, reflected in the negative change in loss upon ablation (darker points in Figure 4c). Interestingly, we find that the projection of this neurons' $\mathbf{w}_{\text{out}}$ onto the frequency direction $\mathbf{v}_{\text{freq}}$ is negative, suggesting that this neuron is suppressing common tokens and promoting rare ones. This indicates that the model is on average biased to predict common tokens even more frequently than their frequency would dictate. This way, lowering confidence by pushing $P_{\text{model}}$ closer to $P_{\text{freq}}$ requires suppressing the token frequency direction, which is the opposite of what one would have naively expected.

## 6 Case study: Induction

As a more detailed case study to examine the active role and confidence-regulating function of entropy neurons, we focus on the setting of induction. Induction occurs when there is a repeated subsequence of tokens in the model's input, and the model must detect the earlier repeat and continue it. Mechanistically, this is implemented by specialized attention heads, called induction heads [16, 57], which attend from some token A to the token B that immediately follows A in the early occurrence and predict B (AB...A → B). Repeated text frequently occurs in natural language (e.g., someone's name) and, during a repeated subsequence, the next token can often be predicted with very high confidence.

### 6.1 Entropy neurons

To analyze this phenomenon, we create input sequences by selecting 100 tokens from C4 [63] and duplicating them to form a 200-token input sequence. Across 100 such sequences, we measure GPT-2 Small's performance and observe a significant decrease in both average loss and entropy during the second occurrence of the sequence (solid lines in Figure 5a). Additionally, we track the activation values of the six entropy neurons in GPT-2 Small analyzed in §3 (dotted lines in Figure 5a). For four of these neurons (11.584, 11.2378, 11.2123, and 11.2910), we note substantial change in average activation values, suggesting that they may play an important role. Furthermore, among final-layer neurons, entropy neurons exhibit the largest change in activation during induction (Appendix H.3, Figure 11).

To further explore the specific effects of entropy neurons on the model's output entropy and loss, we conduct an experiment where, during the second occurrence of the sequence, we clip the activation values of the neurons to their mean values from the first occurrence. The results (Figure 5b) reveal that while the ablation of some neurons such as 11.584 and 11.2123 leads to a slight increase in output entropy, the most significant impact was observed with neuron 11.2378. This results in up to a 70% reduction in entropy, suggesting its role in boosting entropy to counterbalance the model's high confidence during induction. We additionally study the effect of token frequency neurons during induction and we report the results in Appendix H.2, revealing that these neurons also lead to a significant change in entropy compared to randomly selected neurons. We repeat these analyses in naturally occurring induction cases identified in the C4 corpus, obtaining consistent results (Appendix H.1).

## 6.2 Induction head–entropy neuron interaction

In order to verify that induction heads have a causal effect on the activation of our entropy neurons, we perform *BOS ablations* [31]. That is, motivated by the observation that attention heads frequently attend to BOS when inactive (e.g., when induction is not occurring) [72], we set the attention pattern of an attention head to always attend to the first token of the sequence (i.e., BOS) and record the neuron's activation. We perform this procedure on the top 3 induction heads, selected according to the prefix matching score introduced by Olsson et al. [57]. In GPT-2 Small, these heads are L5H1, L5H5, and L6H9, respectively. As baselines, we randomly select 6 heads in layers 5 and 6 that, like our induction heads, have mean attention to BOS > 0.6 but have prefix matching score < 0.1. For both entropy neurons and baselines, we ablate heads individually and in combinations of up to 3 heads.

Figure 5c shows that ablating L5H1 (ind. head 1) leads to a significant decrease in 11.2378's mean activation. Ablating L5H5 (ind. head 2) or L6H9 (ind. head 3) individually does not make a substantial difference, but ablating them alongside L5H1 further decreases activation, bringing 11.2378 close to its activation on the first 100 tokens. We perform additional BOS ablations for other entropy neurons in Appendix H and further study the induction head–entropy neuron interaction in Appendix I, providing preliminary evidence that entropy neurons respond to an internal 'induction-has-occurred' signal that is produced by induction heads.

# 7 Related work

**Uncertainty in language models.**    Uncertainty quantification for machine learning models has been extensively studied to assess the reliability of model predictions [19, 21]. With the increasing adoption of LLMs, accurately determining their confidence has drawn significant attention [22]. Existing research has explored various strategies to calibrate model output, including ensembling model predictions [76, 34], fine-tuning [39, 40], and prompting models to explicitly express their confidence [47, 40, 65, 69]. Additionally, efforts have been made to evaluate the truthfulness [9, 3, 50] and uncertainty [35] of model outputs by inspecting internal representations. However, the investigation of internal, general-purpose mechanisms determining model confidence remains underexplored.

**Interpretability of LLMs' components.**    The analysis of language models' internal components and mechanisms is an increasingly popular area of research (we refer to Ferrando et al. [18] for an overview). Prior work has explored how to attribute and localize model behavior [73, 23, 52], and to identify specific algorithms that models implement to perform tasks [54, 75, 29, 67, 12]. Additionally, many studies have focused on investigating individual neurons [41, 61, 14, 2, 13, 74, 27, 7]. However, neurons are not always the correct unit of analysis, as they can be polysemantic [56, 17]. Recent work has employed sparse autoencoders (SAEs) to find more interpretable linear combinations of neurons [79, 8, 36]. Despite this, in the context of entropy modulation, prior findings of relevant neurons motivated us to focus on neurons.

**Closest related work.**    An anomalous class of neurons characterized by high norm and low composition with the unembedding matrix was observed by Katz and Belinkov [42], though this was not linked to the regulation of output distribution entropy. This class of neurons was independently rediscovered by Gurnee et al. [28] while investigating the universality of neurons in GPT-2 models. Our work connects the concept of entropy neurons to the presence of an effective null space represented by the bottom singular vectors of $\mathbf{W}_U$. Our findings highlight the significance of this seemingly unimportant subspace within the residual stream and align with the observations of Cancedda [10], who associated the bottom $\mathbf{W}_U$ singular vectors with the phenomenon of the attention sink [78].

# 8 Conclusion

This paper presents an analysis of the mechanisms by which LLMs manage and express uncertainty, focusing on two specific components: entropy neurons and token frequency neurons. We show that entropy neurons, with their high weight norm and low direct interaction with the unembedding matrix, affect the model's output via the final LayerNorm. We also introduce and investigate token frequency neurons, which adjust the model's output by aligning it closer to or further from the token frequency distribution. In a case study on induction, we demonstrate the practical implications of these neurons. We show that one example role of entropy neurons is acting as a hedging mechanism to manage confidence in repetitive sequence scenarios. Some limitations of our work include focusing only on two types of components in the neuron basis, relying on proxies for confidence, and observing varying effects across models. We thoroughly discuss the limitations of our work in Appendix A. This study represents the first thorough investigation into the mechanisms that LLMs might use for confidence calibration, providing insights that can guide further research in this area.

## Acknowledgments

This work was conducted as part of the ML Alignment & Theory Scholars (MATS) Program. Throughout the project, we received valuable input from fellow MATS participants. In particular, we would like to thank Andy Arditi, Joseph Bloom, Connor Kissane, and Robert Krzyzanowski. We also extend our gratitude to McKenna Fitzgerald for organizational support. Additionally, we would like to thank Arthur Conmy, Giulia Lanzillotta, Mazda Moayeri, Kumar Shridhar, and Vilém Zouhar for useful discussions and comments on our work. Yonatan was supported by the Israel Science Foundation (grant no. 448/20), an Azrieli Foundation Early Career Faculty Fellowship, and an AI Alignment grant from Open Philanthropy. Alessandro and Ben were supported by a Long-term Future Fund. Alessandro is further supported by armasuisse Science and Technology through a CYD Doctoral Fellowship. Ben is supported though an EPSRC Doctoral Training Partnership Grant. The project was also supported by UK's innovation agency (Innovate UK) grant 10098112 (project name ASIMOV: AI-as-a-Service) and by the European Union under the Horizon Europe vera.ai project (grant 101070093).

## Author Contribution

Alessandro and Ben led the project, carried out the experiments, and wrote the paper. Alessandro proposed the causal formulation and conducted most of the experiments validating the entropy neurons' mechanism of action. He also performed experiments to identify entropy neurons and token frequency neurons across multiple models and wrote the majority of §§ 1 to 5. Ben carried out numerous validation experiments to ensure the robustness of our findings and obtained preliminary results on the universality of entropy neurons. Ben also led the experiments for the induction case study and wrote most of §6 and Apps. H and I. Wes gave frequent and detailed feedback on experiment design and paper write-up. Yonatan, Xingyi, and Mrinmaya provided valuable input on experiment design and presentation of the results. Neel supervised the project and provided guidance at all stages on experiment design and analysis, in addition to editing the paper.

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

## A  Limitations

Our study focuses exclusively on two particular types of final-layer components: entropy neurons and token frequency neurons. However, it is likely that there are multiple other components and mechanisms within language models that contribute to confidence regulation. Exploring these additional components could provide a more comprehensive understanding of how language models manage uncertainty.

We also acknowledge the lack of a precise definition of confidence. Instead, we rely on rough proxies such as entropy and the distance from the token frequency distribution. These measures, while useful, may not fully capture the concept of confidence in language models.

Although we observed the presence of entropy neurons in most of the models we studied, the fraction of their effect explained by LayerNorm re-scaling varies. For instance, in Pythia and Gemma, this fraction is around 40%, which is significantly less than the 80% observed in GPT-2, LLaMA2 7B, and Phi-2. We believe that variations in model architecture and training hyper-parameters may explain these inconsistencies. Although we do not explore these factors in detail, we discovered that the application of dropout during training leads to the emergence of a larger null space (Appendix E), which we hypothesize may encourage the formation of entropy neurons. Future research should rigorously examine the factors that determine the emergence of entropy neurons, as well as other mechanisms that regulate model confidence.

Lastly, our study is primarily focused on how entropy and token frequency neurons contribute to confidence regulation in next-token prediction. An interesting direction for future research would be to investigate confidence-regulating neurons in broader contexts and real-world tasks, such as question-answering or reasoning. This could provide deeper insights into the practical applications and limitations of these neurons in diverse scenarios.

## B  Broader Impacts

This research enhances the understanding of how LLMs regulate confidence in their predictions through entropy neurons and token frequency neurons. While our study primarily advances our technical understanding of language models, some broader impacts should be considered. Enhanced confidence calibration might lead to biased or discriminatory decisions if not adequately mitigated, perpetuating or amplifying existing biases in training data. Additionally, insights from this research could facilitate extraction attacks, compromising data privacy, and be exploited for adversarial attacks, undermining model security. To mitigate these risks, it is essential to integrate robust fairness checks and bias mitigation strategies during model training and deployment. Implementing privacy-preserving techniques, such as differential privacy, can help protect against data leakage. Additionally, developing adversarial defense mechanisms and deploying real-time monitoring systems can counteract potential misuse.

## C  Weight Pre-processing

With the purpose of removing irrelevant components and other parameterization degrees of freedom, we employ a set of standard weight pre-processing techniques [53]. This way, cosine similarities and other weight computations have a mean of 0.

**Folding in Layer Norm.**  Most layer norm implementations include trainable parameters $\gamma \in \mathbb{R}^n$ and $\beta \in \mathbb{R}^n$ (see Eq. (2)). To account for these, we "fold" the layer norm parameters into $\mathbf{W}_{\text{in}}$ by treating the layer norm parameters as equivalent to a linear layer and then combining the adjacent linear layers. We create effective weights as follows:

$$\mathbf{W}_{\text{eff}} = \mathbf{W}_{\text{in}} \cdot \mathbf{diag}(\gamma), \quad \beta_{\text{eff}} = \beta_{\text{in}} + \mathbf{W}_{\text{in}} \cdot \beta.$$

Finally, we center the reading weights because the preceding layer norm projects out the all-ones vector. Thus, we center the weights $\mathbf{W}_{\text{eff}}$ as follows:

$$\mathbf{W}_{\text{eff}}^{'}(i,:) = \mathbf{W}_{\text{eff}}(i,:) - \overline{\mathbf{W}}_{\text{eff}}(i,:).$$

**Writing Weight Centering.**  Every time the model interacts with the residual stream, it applies a LayerNorm first. Therefore, the components of $\mathbf{W}_{\text{out}}$ and $\beta_{\text{out}}$ that lie along the all-ones direction of

the residual stream have no effect on the model's calculations. Consequently, we mean-center $\mathbf{W}_{\text{out}}$ and $\boldsymbol{\beta}_{\text{out}}$ by subtracting the means of the columns of $\mathbf{W}_{\text{out}}$:

$$\mathbf{W}'_{\text{out}}(:, i) = \mathbf{W}_{\text{out}}(:, i) - \overline{\mathbf{W}}_{\text{out}}(:, i).$$

**Unembedding Centering.** Since softmax is translation invariant, we also center $\mathbf{W}_U$:

$$\mathbf{W}'_{\text{U}}(:, i) = \mathbf{W}_{\text{U}}(:, i) - \bar{\mathbf{W}}_{U}(:, i).$$

# D   Experimental details

We carry out all our experiments using data from the C4 Corpus [63], which is publicly available through a ODC-BY license.[9] We use the Pile [20], which is no longer distributed by its creators,[10] solely to compute token frequency statistics, as Pythia was trained on it and we required the original training data for the model. The experiments to measure the total and direct effects, both for entropy and token frequency neurons, were carried out on 256-token input sequences. We used 100 sequences for GPT-2 and Pythia 410M, 50 for Phi-2 and GPT-2 Medium, and 30 for LLaMA2 7B and Gemma 2B. The scatter plot in Figure 3a was obtained on 10000 tokens. The results on the induction case study presented in §6.1 were obtained on 500 input sequences, the shaded area around each line represents the standard error. For the sake of visualization clarity, the box plots in Figure 2 and Figure 3 do not include outliers: the whiskers represent the first and third quartiles. For naturally occurring induction, we selected 1000 sequences from C4 that contained a repeated n-gram (n=6) and met our filtering criteria (each n-gram must be unique and contain no duplicate tokens). The BOS and mean attention output ablation experiments were each carried out on one hundred 100-token input sequences. Baselines for BOS ablation consisted of 6 single-head ablations, 3 two-head ablations, and 3 three-head ablations. Mean attention to BOS (used for baseline selection) was calculated using 1000 sequences from C4. To calculate prefix matching scores for attention heads, we use 1024 sequences of duplicated random tokens.

**Computing resources.** The static analyses of the model weights were conducted on a MacBook Pro with 32GB of memory. The experiments carried out to quantify the neurons' effects were carried out on a single 80GB Nvidia A100. The longest experimental run took approximately 12 hours. Our experiments were carried out using `PyTorch` [59] and the `TransformersLens` library [53]. We performed our data analysis using `NumPy` [30] and `Pandas` [77]. Our figures were made using `Plotly` [37]. The paper's bibliography was curated using `Ryanize-bib` [81].

# E   Unembedding null space & dropout

We hypothesize that the presence of a null space in the model's unembedding matrix $\mathbf{W}_U$ is significantly affected by the application of dropout [66] during training. Dropout represents an incentive for the model to encode information redundantly, aligning it with a specific basis in the residual stream. We expect this redundancy to be reflected in some linear dependencies among the rows of $\mathbf{W}_U$, effectively creating a null space. To empirically validate this hypothesis, we compare the singular values of $\mathbf{W}_U$ between two versions of Pythia 160M [6] – identical except in their application of dropout to the residual stream (Figure 6).

Our observations reveal that the singular values of the model trained with dropout are generally lower, suggesting lower stability and greater linear dependence among the rows of $\mathbf{W}_U$. More notably, the difference in singular values between the two model versions is

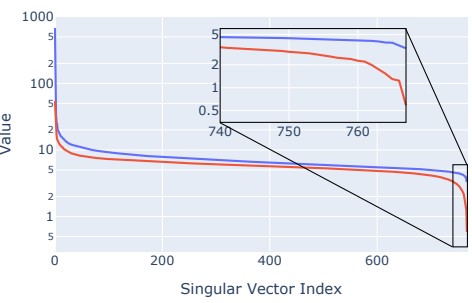

Figure 6: **Effect of Dropout on $\mathbf{W}_U$.** Comparison of singular values for the unembedding matrix between two versions of Pythia 160M—one trained with dropout (red) and one without dropout (blue).

---

[9]`https://opendatacommons.org/licenses/by/1-0/`
[10]`https://the-eye.eu/public/AI/pile/readme.txt`

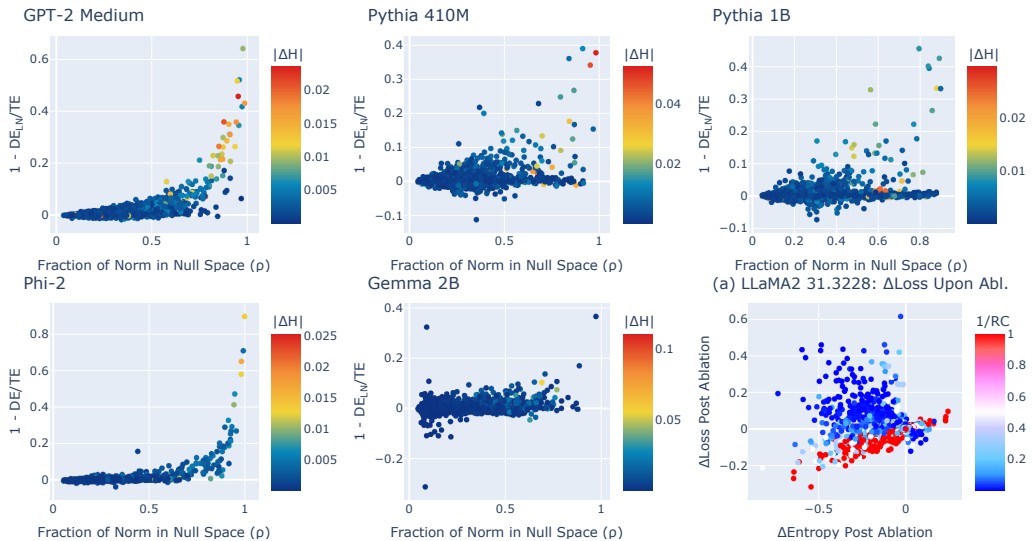

Figure 7: **Entropy Neurons Across Different Models.** Relationship between the fraction $\rho$ of neuron norm in the $\mathbf{W}_U$ null space and the LayerNorm-mediated effect on model output for neurons in the last layer of various models. (a) Change in loss after ablation of entropy neuron 31.3228 in LLaMA2 7B. Ablation decreases entropy and increases loss on incorrect predictions.

larger for the $\sim$10 smallest values, indicating the existence of a subspace in the dropout-trained model that has a substantially reduced impact on the logits. Such findings suggest that entropy neurons are more prevalent in models trained with dropout, as the subspace they leverage within the residual stream is more marked. However, this does not rule out the existence of entropy neurons in models trained without dropout.

## F  Entropy neurons in different models

In Figure 7, we report the results for the final-layer neurons of GPT-2 Medium, Pythia 410M and 1B, Phi-2, and Gemma 2B, complementing our earlier findings from GPT-2 and LLaMA2 7B discussed in §3. $\rho$ is computed by considering the the bottom 1% $\mathbf{W}_U$ singular vectors as null space. We observe a consistent pattern: there exists a subset of neurons that predominantly write to the $\mathbf{W}_U$ null space. However, the extent to which entropy neurons appear in models appears to vary: we observe that entropy neurons in Pythia and Gemma exhibit a weaker effect compared to those in other models. In particular, the fraction of their total effect explained by the LayerNorm scale is around 30-40%, which is notably lower than the approximately 80% observed in GPT-2 Small, LLaMA2 7B, and Phi-2. This suggests that while entropy neurons are a common feature, their strength and the extent of their influence can vary significantly between different models.

### F.1  Additional example of neuron activity

As for neuron 11.2378 in GPT-2 Small in §5, we analyze the strongest entropy neuron in LLaMA2 7B (31.3228). We perform a similar analysis as in §5, where we clip the activation value of the neuron back to its mean when it exceeds it, and we measure the changes in loss and entropy after the ablation (Figure 7a). The ablation of this neuron leads to a significant decrease in entropy, which results in a loss increase on incorrect predictions and a loss decrease on correct predictions.

## G  Token frequency neurons across different models

In Figure 8, we report the results comparing the token frequency-mediated effect against the average absolute change in the KL divergence between $P_{\text{model}}$ and $P_{\text{freq}}$ for neurons at the final layer of GPT-2

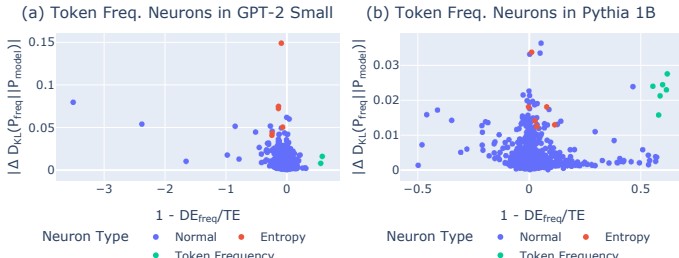

Figure 8: **Token Frequency Neurons Across Different Models.** Relationship between the token frequency-mediated effect and the average absolute change in the KL divergence from $P_{\text{freq}}$ for final-layer neurons in (a) GPT-2 Small and (b) Pythia 1B. Neurons are categorized as normal, entropy, or token frequency neurons.

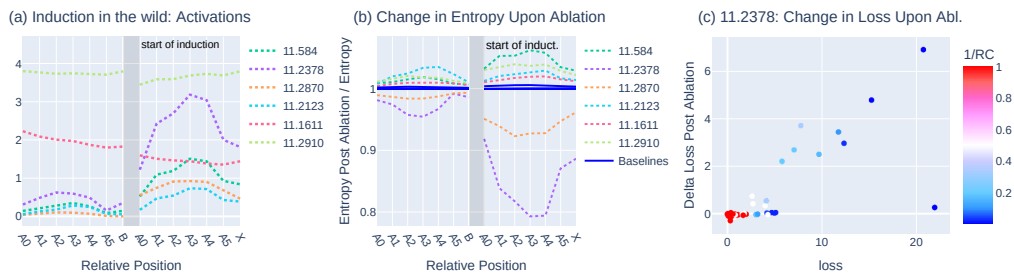

Figure 9: **GPT2-Small Entropy Neurons on Natural Induction**: (a) Change in activation values. (b) Change in entropy upon clipped mean ablation. (c) 11.2389 change in post-ablation loss. Color indicates reciprocal rank of the correct token prediction (1/RC). We use A0-A5 to refer to the positions of the repeated n-gram; B and X indicates the token position that follows the first and second occurrences of the n-gram

Small and Pythia 1B.[11] We also highlight the entropy neurons with the highest LayerNorm-mediated effect, represented by the points with the largest y-axis values in Figure 7.

In both models, we observe a set of neurons with substantial token frequency-mediated effect ($\sim 0.6$). In Pythia 1B, the ablation of these neurons leads to a significant variation in the KL divergence from $P_{\text{freq}}$, comparable to the variations caused by entropy neurons. On the other hand, in GPT-2 Small, the effect of the entropy neurons on $\text{D}_{\text{KL}}(P_{\text{freq}}||P_{\text{model}})$ is significantly larger, suggesting that these neurons might play a more important role on the regulation of the model's output compared to token frequency neurons.

Furthermore, in both models, we observe neurons with significant negative $1 - \text{DE}_{\text{freq}}/\text{TE}$ values, indicating that their effect increases when the impact of the token frequency direction is suppressed. This suggests that the effect through the token frequency direction actually counteracts their effect that bypasses the frequency direction.

## H   Additional Results on Induction

### H.1   Naturally occurring induction

To verify that these neurons behave similarly outside of our artificial setting, we perform the same experiments on examples of induction in natural language data. We select 1000 sequences from the C4 corpus that contain a repeated n-gram of length n=6. In order to remove redundant and trivial examples, we also filter sequences such that the n-gram is unique and no individual n-grams contain duplicate tokens. We perform the same clipped mean ablation procedure as the synthetic setting, mean-ablating to the neuron's activation value on random text.

---

[11]We do not have access to the GPT-2 training corpus. However, since the model's training data included a substantial amount of internet-scraped text, we use a randomly sampled 500M-token subset of the OpenWebText corpus [25] as a proxy for the original distribution.

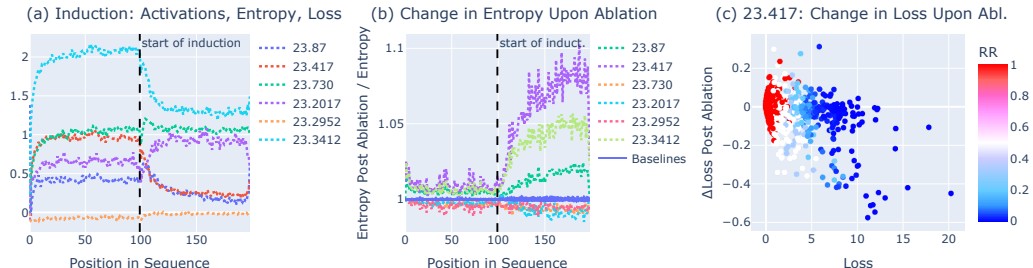

Figure 10: **Token Frequency Neurons on Induction.** (a) Activations, entropy, and loss across duplicated 200-token input sequences. (b) The effect of clip mean-ablation of specific token frequency neurons. (c) Scatter plot of loss changes per token post-ablation of neuron 23.417, colored by reciprocal rank (RR). Ablation tends to increase loss for low initial loss tokens and decrease it for high initial loss tokens.

Figure 9 shows a similar trend to the synthetic setting shown in Figure 5a, though the mean change in activations (and thus change in entropy) is less significant. The most impactful neuron remains 11.2378, the 'hedging' neuron that we believe counter-balances overconfident predictions. Consistent with this hypothesis, when the neuron's effect is removed via (clip) mean ablation, the model's predictions become more confident: entropy decreases by up to 20% of its original value.

The other entropy-increasing neuron shown in Figure 9b is 11.2870. Unlike the other 4 entropy neurons, it fires strongly only on the first few tokens of induction and then decreases in activation (Figure 12). In both synthetic and natural induction settings, 11.2870 has an entropy-increasing effect (Figure 5b and Figure 9b), suggesting that it increases uncertainty at the start of induction where the model should be less certain that copying previous tokens will lead to the correct next-token prediction. (As an example, for the phrase 'Mr. Smith ... Mr.', the model should not be overly confident in predicting that the next token will be 'Smith', since a different name could follow instead.) Since we choose a relatively short prefix length for our natural induction setting (n=6), the neuron is active across our entire repeated sequence and increases entropy.

## H.2 Token frequency neurons

To analyze the activity of token frequency neurons in the context of induction, we repeat the analysis carried out for entropy neurons. Using 100 input sequences created by concatenating two repetitions of a 100-token sequence, we study the activation values (averaged across token positions) for six token frequency neurons identified in §4 within the last-layer MLP in Pythia 410M. We observe that, in this setting, four neurons (417, 3412, 2017, and 87) display a substantial increase in activation value (Figure 10a). Upon clipped mean-ablation, neurons 417, 3412, and 87 lead to a significant increase in entropy compared to other neurons and a baseline of 10 randomly selected neurons (Figure 10b).

Focusing more closely on neuron 417, we find that reversing its state change leads to a somewhat positive delta in loss for tokens with initially low loss, and larger, negative changes in loss for tokens with high initial loss (Figure 10c). This trend, which is inverse to that observed with entropy neuron 2378 in GPT-2, demonstrates how different mechanisms of entropy modulation can operate at different levels to balance the model's output distribution.

## H.3 Change in activation of final-layer neurons

Figure 11 shows that entropy neurons exhibit among the largest changes in activation for final-layer neurons in GPT2 Small and LlaMa2 7B, suggesting that they are performing an important function in induction scenarios. When comparing synthetic and natural induction, we note that the set of most-changing entropy neurons is different (but overlapping). This is explained by the fact that most entropy neurons increase in activation over the course of induction (e.g. 11.2378), whereas others, such as 11.2870 only fire strongly at the start of induction. As discussed in Appendix H.1, our natural induction setting highlights start-of-induction neurons, since it consists of short repeated n-grams (n=6). By contrast, our synthetic setting consists of long 100-token repeated sequences and so favors neurons that continuously increase in activation.

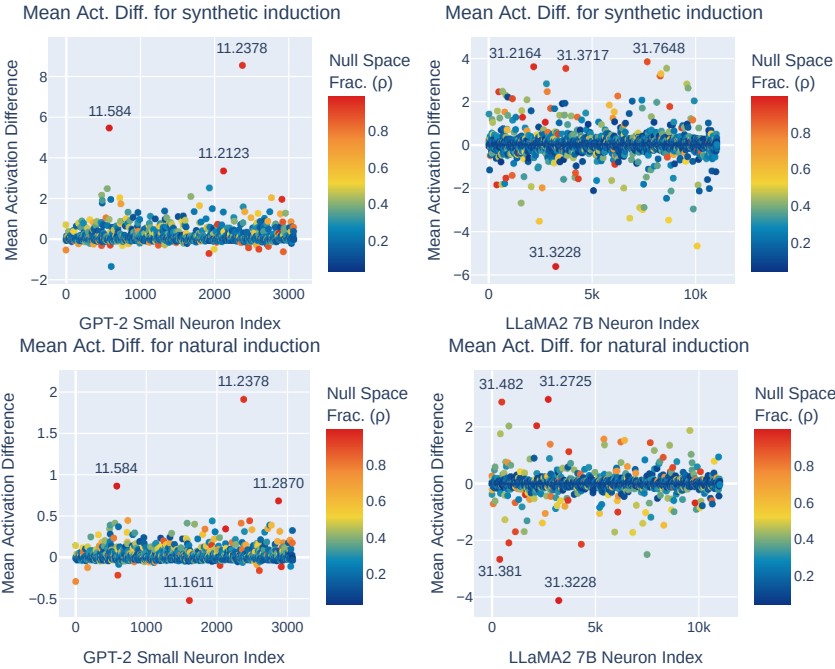

Figure 11: **Neuron activation differences for synthetic and natural induction**: (left) GPT2-Small final-layer neurons (right) LLaMa2 7B final-layer neurons. Among final-layer neurons, entropy neurons exhibit the largest activation change for both synthetic and natural induction settings. Entropy neurons with especially high fraction of neuron norm in the null space ($\rho$) and large change in activation are labelled.

# I   Induction head–entropy neuron interaction

In this section, we study the interactions between induction heads and entropy neurons in more detail. In Appendix I.1 we use BOS ablations to verify that induction heads have a significant causal effect for 5/6 of our entropy neurons. Next, in Appendix I.2 we provide preliminary evidence that induction heads produce an internal signal that indicates when induction has occurred, and that our entropy neurons respond to this signal. Finally, in Appendix I.3 we check the direct composition between entropy neurons and induction heads, showing that induction heads do not have especially strong direct effect, and identifying a set of components that may mediate the 'induction-has-occurred' signal.

## I.1   BOS Ablations for GPT2 Small neurons

We present the full results of BOS ablations (§6.2) for all six of the entropy neurons under investigation in GPT2 Small (Figure 12). Neurons 11.584, 11.2123, and 11.2910 display a similar trend to 11.2378: intervening on the top induction head, L5H1, leads to a substantial decrease in activation, suggesting it has an important causal effect. By comparison, ablating baseline heads in layers 5 and 6 have little impact on activation.

We note that for 11.2870, our start-of-induction neuron (Appendix H.1), BOS ablation of L5H1 has a strong causal effect, but that the direction of this effect changes: on the first few tokens of the repeated sequence, the head boosts activation (activation decreases upon BOS ablation) but it suppresses activation across the remainder of the sequence (activation increases upon ablation). Prior work has found that attention head L5H1 is specialized for long-prefix induction (induction over long sequences) [44]. Thus, we believe 11.2870 is a concrete example of where such specialization is useful: the output of L5H1 enables 11.2870 to fire strongly (and increase entropy) only at the start of induction.

## I.2   Mean attention output ablations

Because entropy neurons respond strongly to induction and are causally influenced by induction heads, we hypothesize that induction heads produce a generic 'induction has occurred' signal that

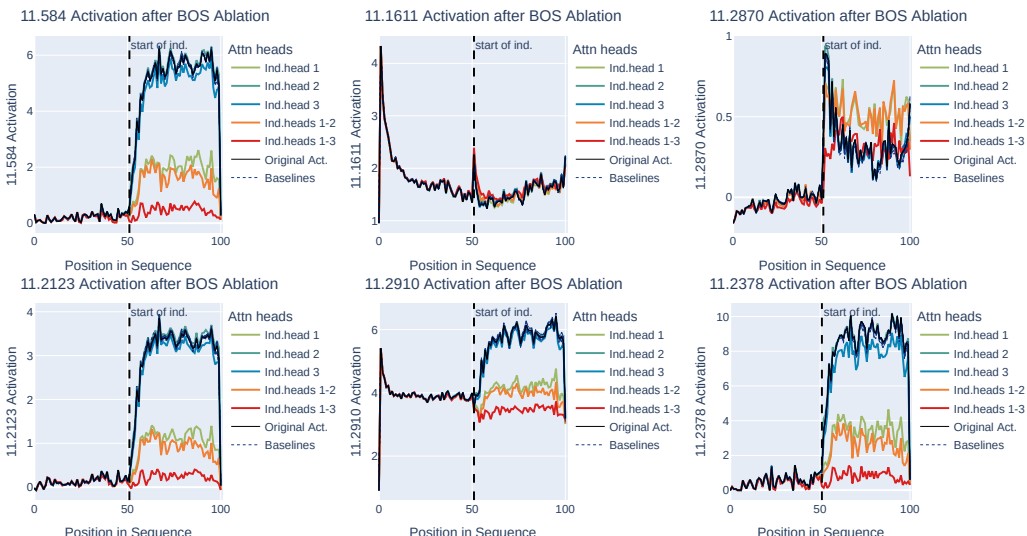

Figure 12: **GPT2-Small Entropy Neurons after BOS Ablations**: Ablation of induction heads leads to substantial decrease in activation for most entropy neurons.

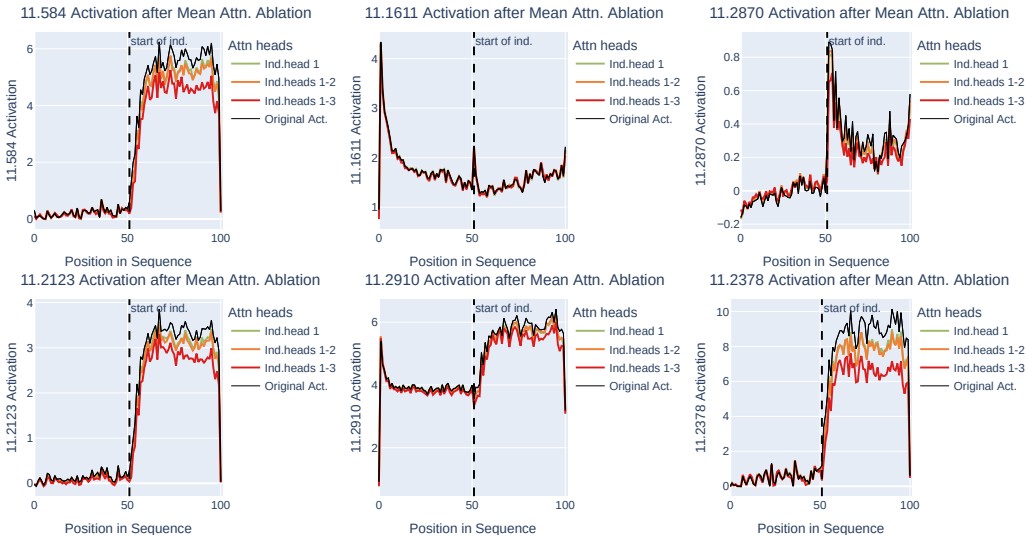

Figure 13: **GPT2 Small Entropy Neurons after Mean Attention Ablations**: Replacing induction head outputs with their mean induction output leads to similar activations.

indicates to later components that the model is in an induction context. To test this hypothesis, in our synthetic induction setting, we perform mean attention output ablations: we replace each induction head's contribution to the residual stream with its mean contribution averaged over the second half of the sequence (i.e. when induction is occurring).[12] In this way, we hope to retain the generic "induction has occurred" signal, while removing token-specific behavior, such as directly boosting the logits of an attended-to token.

Figure 13 shows that mean attention ablations are able to produce activations that are close to their original values. For example, for 11.2378, mean attention ablation of the top induction head results in small activation decrease from $\sim$9 to $\sim$8. For comparison, recall that BOS ablation of this same head causes mean activation to fall to roughly 4. This suggests that a substantial portion of the

---

[12]In practice, this is implemented using attention **Z** vectors ('attention output pre-linear') rather than attention output. However, the two are equivalent for our purposes since **Z** vectors are converted to attention outputs by a linear map [53]

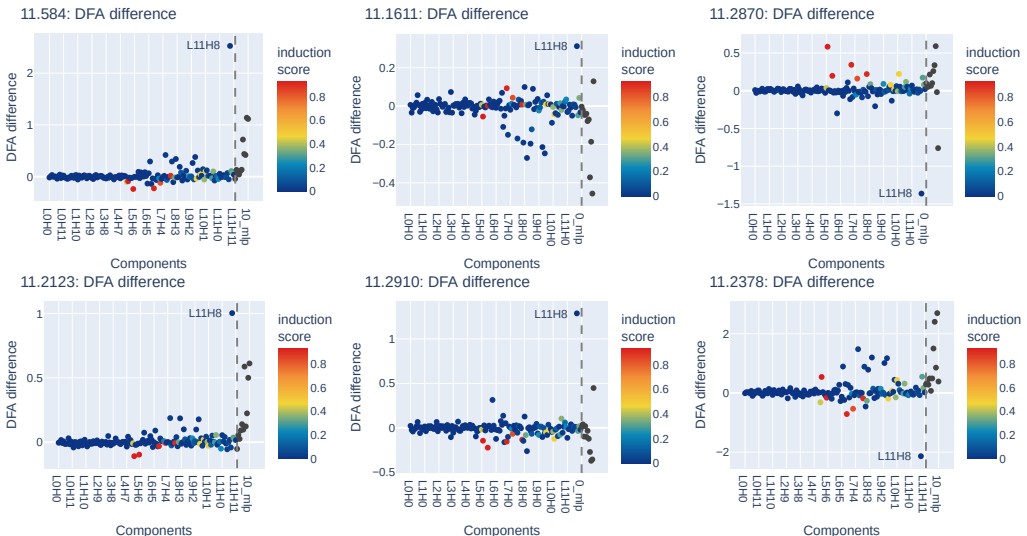

Figure 14: **GPT2 Small Direct Effect of attention heads and mlps on induction**: DFA difference between induction and non-induction sequences. Attention heads are to the left of the dotted line. Induction score refers to the prefix matching score.

entropy neurons' activation is derived from this generic induction signal, rather than token-specific information.

We note that though mean attention ablation is sufficient to raise the activations of our entropy neurons close to their original activations on the second half of the sequence, they do not have much impact over the first half of the sequence. (We would expect activation to increase, since we are adding an 'induction is happening' signal.) We speculate that this is because there are other important components that we have not factored into our analysis. For example, the signal from induction heads to entropy neurons is likely mediated by later-layer MLPs/Attention Heads. These could perform an AND operation over our induction heads' signals and duplicate token heads, such that both signals are necessary in order to produce the trigger for our entropy neurons. We leave a more thorough investigation of this potential circuit to future work.

### I.3 Direct effect of components

In this section, we study whether the output of induction heads directly contributes to the activation of our entropy neurons, or whether their effect is mediated by other components. To do this, we produce a corrupted version of our induction sequences by replacing the first 100 tokens of each sequence with another random 100-token sequence sampled from C4. Then, for both our original and corrupted sequences, we use direct feature attribution (DFA) [43] to calculate the direct contribution of each preceding component in the transformer to our neuron's activation across the last 100 tokens. This is done by taking the dot product of the output of each component (attention heads and MLPs) with the neuron's $W_{in}$ vector, after applying LayerNorm.

Figure 14 shows the difference in DFA between the original (induction) sequence and the corrupted (non-induction) sequence, indicating which components were responsible for the change in activation. For entropy neuron 11.2870 induction heads have a strong direct contribution to the neuron's activation. However, this is not the case for our other 5 entropy neurons. In particular, L11H8 as well as MLPs in layers 8-10 have a strong direct effect, suggesting that the 'induction has occurred' signal effect may be mediated by these other components. Further evidence for this is suggested by the observation that the direct effect of induction heads is often opposite to the expected change in activation. That is to say, for entropy neurons that increase in activation upon induction, the direct effect of the induction heads is often negative (i.e. suppressing activation), and becomes even more negative upon induction. Since our BOS ablation experiments show that turning off these heads leads to a *decrease* in activation, this suggests the presence of intermediate components that convert the negative direct effect signal from induction heads into a positive one.

