# OpenReview forum: "Confidence Regulation Neurons in Language Models"
_NeurIPS.cc/2024/Conference — NeurIPS 2024 poster_

### Official Review · Reviewer_S9nW · 2024-07-03

**Soundness:** 4
**Presentation:** 4
**Contribution:** 4
**Rating:** 9
**Confidence:** 5

**Summary:**

This paper studies how large language models (LLMs) regulate uncertainty in next-token predictions through specific components: entropy neurons and token frequency neurons. Entropy neurons, identified by their high weight norm and minimal direct impact on logits, influence model confidence by operating within an unembedding null space. Token frequency neurons adjust logits based on token frequency, modulating the output distribution toward or away from the unigram distribution. The study includes a detailed examination of these neurons in various models, demonstrating their role in managing prediction confidence, particularly in repeated sequence scenarios.

**Strengths:**

- The extensive experimental validation, including ablation studies and cross-model analysis, gives empirical support to the theoretical claims, enhancing the overall robustness and reliability of the study.
- The paper is very well written, and it was easy to follow through even though the topic is rather complex.
- I think that the paper provides an good mechanistic explanation of how entropy neurons operate through the unembedding null space and LayerNorm, improving our understanding of their indirect impact on model predictions.

Overall, I find the paper's findings very valuable, the methodology is rouboust, and the authors adressed themselves the few limitations of the paper. I honestly enojoyed reviewing this paper.

**Weaknesses:**

- As the authors say in the "limitations" section, they use entropy and distance from the token frequency distribution as proxies for confidence, which may not fully capture the complexity of confidence in language models.
- The extent to which entropy neurons influence model output via LayerNorm seems to change across models. This variability suggests that model architecture and training parameters could play a important role.

**Questions:**

Please, read my questions and suggestion as a starting point for a follow-up paper.
- Could you provide a more precise definition of confidence in the context of LLMs?
- While LayerNorm plays a crucial role in the functioning of entropy neurons, what might be the impact of other normalization techniques (for example, BatchNorm or RMSNorm) on these neurons?
- I'd suggest also to perform an analysis of how specific features of the models and training hyperparameters influence the effectiveness of entropy and token frequency neurons.

**Limitations:**

The authors properly addressed all the possible limitations of the paper.

---

> ### Author Rebuttal · Authors · 2024-08-07
>
> Thank you for recognizing that our work provides a “good mechanistic explanation of how entropy neurons operate [...], improving our understanding of their indirect impact on model predictions.” We are encouraged by your recognition of our findings as "very valuable" and appreciate that you enjoyed reviewing our paper. Your positive feedback is greatly appreciated.
>
> > **Weakness 1** & **Q1:** Could you provide a more precise definition of confidence in the context of LLMs?
>
> This is a good and challenging question. Defining and quantifying uncertainty in machine learning models, including LLMs, is a complex challenge. In general, the uncertainty of a machine learning model can be divided into two types: aleatoric, which arises from inherent noise in the data, and epistemic, which stems from the model’s lack of knowledge [1].
> In the context of LLMs, which predict a probability distribution over a set of possible next tokens, possible candidates for gauging the model's confidence are, for instance, the largest predicted probability value, the gap between the two highest probabilities, or other, more global properties of the output distribution, such as entropy or divergence from a baseline distribution [2, 3]. We opt for the latter measures as single probability scores neglect broader characteristics of the output distribution.
> In particular, the token frequency distribution represents a natural baseline for next-token prediction, which models resort to at early stages of training [4, 5], and serves as an educated guess based on the previously seen tokens.
> Although we believe that our chosen proxies are reasonable and practical measures for assessing a model’s confidence in its next-token prediction, we acknowledge the challenges in uncertainty estimation. We will add a note to the paper to highlight this point.
>
>
> > **Q2:** While LayerNorm plays a crucial role in the functioning of entropy neurons, what might be the impact of other normalization techniques (for example, BatchNorm or RMSNorm) on these neurons?
>
> RMSNorm is essentially identical to LayerNorm, except it does not center the residual stream by subtracting its mean. Since the impact of entropy neurons is mediated through the re-scaling operation of LayerNorm rather than the centering, the function and presence of entropy neurons should not be affected by using RMSNorm instead of LayerNorm. Empirical validation of this comes from LLaMA 2, which uses RMSNorm and still exhibits the presence of entropy neurons.
> BatchNorm, on the other hand, normalizes along the batch dimension. This operation involves normalizing across a dimension external to the model's architecture because the model, during a forward pass on input $x$, has no access to information about activations computed on other inputs $x'$ in the same batch. Therefore, we do not expect BatchNorm to impact the presence or function of entropy neurons.
>
>
> > **Q3:** I'd suggest also to perform an analysis of how specific features of the models and training hyperparameters influence the effectiveness of entropy and token frequency neurons.
>
> This is definitely an interesting question. In Appendix E, we analyze the effect of one training hyperparameter: the application of dropout during training. We study this in two versions of Pythia 160M that differ only in the application of dropout. We observe that dropout leads to lower unembedding singular values and, interestingly, to a more pronounced effective null space, making the presence of entropy neurons more pronounced. We agree that a more systematic analysis of the impact of architectural choices on confidence regulation neurons is an interesting avenue for future research and could provide valuable insights into how these mechanisms emerge.
>
>
> ---
>
> [1] Kendall, A. and Gal, Y., What uncertainties do we need in bayesian deep learning for computer vision?. NeurIPS 2017.
> [2] Huang, Y., et al., Look before you leap: An exploratory study of uncertainty measurement for large language models. arXiv 2023.
> [3] Yoshikawa, H. and Okazaki, N., Selective-LAMA: Selective Prediction for Confidence-Aware Evaluation of Language Models. In Findings of EACL 2023.
> [4] Meister, C., et al., A Natural Bias for Language Generation Models. ACL 2023.
> [5] Chang, T.A. and Bergen, B.K., Word acquisition in neural language models. TACL 2022.

---

> > ### Comment · Reviewer_S9nW · 2024-08-10
> >
> > Thank you for your detailed responses, I appreciate the clarifications provided, particularly regarding the definition of confidence in LLMs and the impact of different normalization techniques on entropy neurons.

---

### Official Review · Reviewer_Thsu · 2024-07-10

**Soundness:** 3
**Presentation:** 3
**Contribution:** 2
**Rating:** 5
**Confidence:** 4

**Summary:**

The paper investigates specific neurons in LLMs  (termed "confidence regulation neurons")  that modulate the uncertainty of the next token prediction by modulating the output distribution. First, entropy neurons modulate the overall entropy of the output distribution by writing to an effective null space of the unembedding matrix, thereby influencing the residual stream norm with a minimal direct effect on the logits themselves. Second, token frequency neurons modulate the predictive distribution proportionally to the token frequency in the training data.

**Strengths:**

- The paper clearly written and well-organized, making complex concepts accessible.
- It provides deeper insight into the role of entropy neurons in regulating the confidence of LLMs through the unembedding null space.
- It introduces token frequency neurons, a type of neurons that have not been discussed in prior work.
- It demonstrates the presence of these entropy and token frequency neurons across various models.

**Weaknesses:**

- **Novelty**: The authors claim that they provide a "novel example of how language models can use LayerNorm to indirectly manipulate the logit values" and that prior studies "fail to account for the normalization effects imposed by LayerNorm" (lines 160 to 163).
However, the mechanisms of entropy neurons have already been discovered in previous work [1]. For instance, Gurnee et al. (2023) show that entropy neurons "modulate the model’s uncertainty over the next token by using the layer norm to squeeze the logit distribution, in a manner quite similar to manually increasing the temperature when performing inference.". Although prior studies do not differentiate between the total and direct effect of an entropy neuron on the output distribution, the novelty of the analysis is not clear-cut.

- **Clarity**: Individual neurons are referred to as *layer.position* and simply *position* interchangebly (e.g. 11.2378 and 2378). Since only neurons in the final layer were investigated, referring to the layer is redundant. Also, the authors should consider further simplifying the name of the neurons, as the exact *position* does not add immediate value to the reader.

- **Interpretability**: The interpretation of the results is sometimes unclear. For instance, the authors observe that token frequency neurons suppress common tokens and promote rare ones. They suggest this indicates the model is biased to predict common tokens more frequently than their actual occurrence. However, an alternative interpretation could be considered. The model might also assign high probability to a single rare token while assigning almost no probability to other rare tokens. Promoting all other rare tokens to match their token frequency would bring the output distribution closer to the token frequency distribution, while increasing entropy and lowering confidence. In other words, the model's bias might not be just towards common tokens but also towards certain rare tokens. Such alternative interpretations would also resolve the counterintuitive explanation. The authors should provide more insights to support their interpretations.

---
[1] Wes Gurnee, Theo Horsley, Zifan Carl Guo, Tara Rezaei Kheirkhah, Qinyi Sun, Will Hathaway, Neel Nanda, and Dimitris Bertsimas. Universal neurons in gpt2 language models.

**Questions:**

- Why did the authors analyze 6 entropy neurons but only 5 token frequency neurons?
- In Figure 5(a), are entropy, loss, and neuron activations really all on the same scale (single y-axis)?
- Why does the induction case study not consider the novel token frequency neurons but only the known entropy neurons?

**Limitations:**

The authors adequately addressed limitations.

---

> ### Author Rebuttal · Authors · 2024-08-07
>
> Thank you for recognizing that our work “provides deeper insight into the role of entropy neurons” and that our paper is “clearly written and well-organized, making complex concepts accessible”.
> > **Novelty**: The novelty of the analysis is not clear-cut.
>
> Gurnee et al. identified "entropy neurons" with large norms and low compositionality with the unembedding matrix across GPT-2 models trained with different seeds. They hypothesized that these neurons modulate the model's uncertainty through LayerNorm. Their experiments involved artificially increasing these neurons' activation values, leading to significant changes in LayerNorm scale and entropy. While insightful, these observations do not conclusively establish LayerNorm as the primary mechanism.
> Our work extends these findings in several ways:
> 1. We identify and quantify the presence of an unembedding null space.
> 1. We differentiate and quantify the total and direct effects of entropy neurons, establishing LayerNorm as the key mechanism mediating their impact.
> 1. We analyze examples where entropy neurons naturally achieve maximum activation, providing insights into their behavior.
> 1. We detect entropy neurons across multiple models, reinforcing the generality of our findings.
> 1. We demonstrate the practical implications of entropy neurons in the induction setting, highlighting their role in modulating model confidence.
>
> In conclusion, our study offers a substantially deeper and more rigorous investigation into the mechanisms of entropy neurons, building on the work of Gurnee et al. while introducing novel analyses and findings. Combined with our discovery of token frequency neurons, we believe our work provides a substantial degree of novelty and contributes meaningfully to the understanding of confidence regulation in language models.
> > **Clarity:** Individual neurons are referred to as layer.position and simply position interchangeably
>
> Thank you for pointing this out. We will add a sentence to clarify the notation and ensure consistency across the entire paper.
> > **Interpretability:** The model might also assign high probability to a single rare token while assigning almost no probability to other rare tokens.
>
> Thank you for helping us consider alternative hypotheses for our observations. It is important to explore different interpretations, and we appreciate your feedback.
>
> To clarify this point, we dug deeper into the effect of token frequency neurons on the model output. We know that token frequency neurons suppress common tokens and promote rare ones, and the fact that we observe an increase in entropy upon such contribution leads us to conclude that (1) the model's output distribution is typically skewed towards assigning probability mass to common tokens that is higher than the token frequency dictates. The reviewer suggests that the same change in entropy could be observed if (2) the model assigns high probability to a single rare token while assigning almost no probability to other rare tokens.
>
> To test these interpretations, we studied the change in the KL divergence between the model's output distribution (averaged over 15M tokens) and the token frequency distribution, while adding to the model’s output logits the logits representing the unigram distribution (multiplied by a factor $k$ that we vary). Adding this vector promotes common tokens and suppresses rare ones, while subtracting it achieves the opposite effect, simulating the contribution of token frequency neurons.
>
> In the scenario suggested by Interpretation 2, adding the token frequency vector to the output logits would remove probability mass from the rare tokens (and particularly from the rare token that received a high probability score) and increase the probability assigned to common tokens. If this were the case, we would observe a final output distribution more aligned with the token frequency distribution, indicated by a decrease in the KL divergence between the two distributions.
>
> According to Interpretation 1, when adding the token frequency vector to the output logits, the already high probability assigned to common tokens should become even larger, making the output distribution even more skewed. Therefore, the KL divergence between the model's output and the token frequency distribution should grow even larger.
>
> The results of our analysis are illustrated in the attached PDF. We observed that the KL divergence between the model's output and the token frequency distribution increases as we add the token frequency vector to the logits, which is consistent with Interpretation 1. We will include these results in the final version of the paper.
> > **Q1:** Why did the authors analyze 6 entropy neurons but only 5 token frequency neurons?
>
> There was no particular reason for analyzing 6 entropy neurons but only 5 token frequency neurons. Our results (Figs. 2e, 2f, and 3b) show that entropy neurons and token frequency neurons exist on a continuous spectrum rather than distinct clusters. For our in-depth analyses, we focused on the most pronounced outliers to represent these mechanisms effectively, rather than investigating every neuron that exhibited these characteristics.
> > **Q2:** In Figure 5(a), are entropy, loss, and neuron activations really all on the same scale?
>
> Correct, they are all on the same scale.
> > **Q3:** Why does the induction case study not consider the novel token frequency neurons but only the known entropy neurons?
>
> We do provide results for token frequency neurons on induction in Appendix H. Similar to entropy neurons, the activations of token frequency neurons change substantially upon induction.
>
> ---
> In conclusion, we would like to thank the reviewer for their feedback, particularly regarding the interpretation of our results. We will include the additional analysis in the paper to provide further clarity. We hope that, in light of these clarifications and additional analysis, the reviewer will consider increasing the overall rating.

---

> ### Author Response · Authors · 2024-08-12
>
> Before this phase of the discussion period ends, we wanted to check in with the reviewer on whether we have addressed your concerns with our work?

---

> > ### Comment · Reviewer_Thsu · 2024-08-13
> >
> > Thank you for the rebuttal; it addressed the majority of my concerns and questions. My current score reflects my overall assessment of this work pretty well, so I have decided to maintain the score.

---

### Official Review · Reviewer_BDF8 · 2024-07-11

**Soundness:** 3
**Presentation:** 3
**Contribution:** 3
**Rating:** 7
**Confidence:** 3

**Summary:**

This paper investigates two kinds of neurons by which transformer language models calibrate their predictions. These are (1) “entropy neurons”, which can affect logit values, but do not promote specific tokens, and (2) “token frequency neurons”, which influence a model’s likelihood of outputting bigram word statistics.

**Strengths:**

- Calibration of model confidence is an important area of study for trustworthy deployment of ML systems.
- The experiments attempt to show that both kinds of neurons for model calibration arise across various model sizes and architectures up to LlaMa-2-7B, suggesting this description is general to an extent (varying success across models).
- The results have nice synergy with existing work, providing a clearer understanding of how these previously discovered neurons can influence model behavior.

**Weaknesses:**

- It is unclear what the connection is between “token frequency neurons” and “entropy neurons”. Do these two kinds of neurons interact (if so, how?) or are they separate mechanisms by which models calibrate logits? As this is a major part of the paper, it should be made clear.

- The case studies provided are pretty simple settings. Studying the effect of calibration neurons on more real-world tasks like question answering - where there is a small set of potential answers and seeing if and how the same mechanisms apply would help strengthen the generality of the claims made. (This is noted by the authors in the limitations section, and I agree with them).

**Questions:**

- The total effect is much larger than the direct effect for both sets of neurons. For the “token frequency neurons”, does this mean they are influencing other components that actually promote bigram statistics? Or what exactly is their effect mediated by? It is unclear to me how the same direction could promote the most common bigrams for every distinct token in the vocabulary.

- Are entropy neurons and token frequency neurons only studied at the last layer of the model? Do they appear elsewhere, and are just strongest at the last layer?

- Does the discovery of these neurons influence how we should think about using direct logic attribution as a way to understand components and their interactions? Lines 70-71 briefly mention this, but could you can briefly expand on what is meant here?

**Limitations:**

The authors do point out a number of limitations to their study, which are valid and could strengthen the paper when/if addressed. Though many may not be in scope.

---

> ### Author Rebuttal · Authors · 2024-08-07
>
> Thank you for recognizing that our work addresses “an important area of study for trustworthy deployment of ML systems”, providing “a clearer understanding of how these previously discovered neurons can influence model behavior.”
>
> > **Weakness 1:** Do these two kinds of neurons interact (if so, how?) or are they separate mechanisms by which models calibrate logits?
>
> The two types of neurons represent separate mechanisms that language models can implement to calibrate their logit values. Their activation might be triggered by different features of the input, and they affect the output distribution differently. While they can be active simultaneously (e.g., during induction, where entropy neurons play the most significant role), they exhibit distinct activation patterns. Token frequency neurons tend to activate densely, with activation values significantly larger than 0 occurring frequently. In contrast, entropy neurons generally activate sparsely, often in response to specific sequences like repetitions, structured strings (e.g., email addresses and links), and common phrases or n-grams that are likely repeated in the training data.
>
> > **Weakness 2:** Studying the effect of calibration neurons on more real-world tasks like question answering [...] would help strengthen the generality of the claims made. (This is noted by the authors in the limitations section, and I agree with them).
>
> We agree with the reviewer that showing the effect of entropy and token frequency neurons on more real-world tasks like question answering would provide strong evidence for the generality of the mechanisms we study. However, we would like to highlight that our research is the first to investigate these specific mechanisms within language models. By identifying and characterizing entropy neurons and token frequency neurons, we provide a foundational understanding that future work can build upon. Our findings offer an important starting point for exploring these mechanisms in more complex and varied settings, and we are optimistic that subsequent research will extend our work to a broader range of tasks, including real-world applications like question answering.
>
> > **Q1:** The total effect is much larger than the direct effect for both sets of neurons. For the “token frequency neurons”, does this mean they are influencing other components that actually promote bigram statistics? Or what exactly is their effect mediated by? It is unclear to me how the same direction could promote the most common bigrams for every distinct token in the vocabulary.
>
> Token frequency neurons work by promoting or suppressing the *unigram* distribution component in the model's output. This means they adjust the model's predictions based on the individual token frequencies rather than sequences of two tokens (bigrams). We demonstrate this mechanism by identifying the residual-stream direction representing the unigram/token frequency distribution. (To provide some intuition: increasing the value of the residual stream along this direction promotes common tokens and suppresses uncommon ones.) We show that this direction mediates a significant portion of these neurons’ effect on the output: when we prevent the residual stream from varying along this direction (isolating the direct path that "bypasses" the token frequency direction), the effect of these neurons decreases significantly. In other words, the total effect is larger than their direct effect (which excludes the contribution along the token frequency direction), indicating that these neurons significantly influence the model's output distribution by modifying the residual stream along the unigram/token frequency direction.
>
> > **Q2:** Are entropy neurons and token frequency neurons only studied at the last layer of the model? Do they appear elsewhere, and are just strongest at the last layer?
>
> We focused our study on final-layer neurons, as we expect entropy neurons to be strongest at the final layer, where their effect on the output cannot be mediated by other intermediate model components. However, these neurons might also be present in previous layers. An exhaustive search and comparison of entropy neurons across layers represent an interesting direction for future analysis.
>
>
> > **Q3:** Does the discovery of these neurons influence how we should think about using direct logic attribution as a way to understand components and their interactions?
>
> Thank you for bringing up this point. We believe this is an interesting insight from our study. Interpretability analyses based on direct logit attribution typically involve projecting an internal model representation or weight vector onto the vocabulary space to determine whether the set of tokens being promoted share a specific feature or concept [1, 2]. Our work shows that entropy neurons write onto a residual-stream effective null space, which gets mapped onto a neighborhood of the zero vector in the vocabulary space. If we were to interpret these neurons via direct logit attribution, we would mistakenly conclude that their impact on the model prediction is minimal and uninterpretable, missing their interaction with the final LayerNorm.
> These observations suggest that, while performing direct logit attribution, we must consider that some subspaces in the model residual stream might not influence the next token prediction directly. Thus, we might be trying to understand the *direct* contribution of a model component to the final prediction when its main effect is *indirect* (e.g., mediated by LayerNorm) and its direct effect is minimal, even when the component studied is at the final layer of the model.
>
> ---
>
> [1] Elhage, N.,et al., A mathematical framework for transformer circuits. Transformer Circuits Thread, 2021.
> [2] Geva, M., et al., Transformer Feed-Forward Layers Build Predictions by Promoting Concepts in the Vocabulary Space. EMNLP 2022.

---

> > ### Comment · Reviewer_BDF8 · 2024-08-08
> > **Thank you for the reply**
> >
> > Thank you for the detailed reply. After reading the authors' rebuttals and other reviews, all of my questions/concerns have now been addressed and I think the paper would be a solid contribution to the conference. As such, I have raised my score to a 7.

---

> > > ### Author Response · Authors · 2024-08-09
> > >
> > > We sincerely appreciate your positive feedback. Thank you!

---

### Official Review · Reviewer_C4Wm · 2024-07-13

**Soundness:** 3
**Presentation:** 3
**Contribution:** 2
**Rating:** 7
**Confidence:** 3

**Summary:**

This work studies a small but potentially important subset of neurons in the final layer for a trained transformer model that appear to regulate the confidence of a model (proxied by the variation in entropy of the model's output). Specifically, the paper builds upon entropy neurons found in previous work, extending the analysis further to show that these entropy neurons (which have high norm but low composition with the unembedding matrix) essential modify the null space of the unembedding matrix. This effectively scales the logits without changing the relative ranks of the tokens, thus regulating confidence. The paper continues this exploration by finding "token frequency" neurons, which are neurons that move the model's output distribution towards unigram token frequency distribution. The work claims that this is another form of confidence regulation, essentially "moving" the distribution towards unigram frequencies when the model is not confident about its prediction.

The paper provides the methodology for finding these neurons, as well as performs ablation/intervention experiments on these neurons to measure the difference in "confidence". Experiments are conducted on several pre-trained models, including GPT2, Pythia, LLaMA-2, Gemma and Phi-2. The paper also ends with a specific analysis of "induction"-behavior depicted in recent mechanistic interpretability works, studying how entropy/token-frequency neurons affect the output distribution when sequences are repeated in the output.

**Strengths:**

- The overall paper is well written, describing its motivation well, as well as the methodology. The figures presented do a good job at compressing the useful information in an accessible form.
- The experiments conducted as convincing, and improve upon the findings of the previous works on which this paper is built upon. The experiments are also conducted on several models, which further showcases the universality of the findings.
- The paper focuses on an important problem of understanding transformer models (although the immediate practical applicability of the findings is limited, it still answers important questions)

**Weaknesses:**

- While the results presented in the paper are indeed exciting, they raise a lot of questions (see Questions section) that are easy to answer but have been ignored by the paper. For instance, while the paper has run experiments on many models, there is very little discussion of these results. The paper would be much stronger if some additional discussion revolving around model and vocabulary sizes was included.
- The paper mentions a few existing works on confidence regulation and calibration, but does not tie the work done in this paper within the context of this broader field.

**Questions:**

- How are the token frequencies computed for the second half of the paper? Are these sub-word frequencies? On what dataset are these computed, and do you expect the training data distribution to have any effect on the results?
- How many entropy neurons usually exist? From the plots, it seems this number may be somewhere in the single digit range? Explicitly stating this will still be useful information for the reader
- Can you hypothesize why Gemma 2B has few to no entropy neurons?
- Relatedly, what is the effect of model size on the number of these entropy/token-frequency neurons? Should one assume that model size has no effect? What about the vocabulary size?
- Token freq is confusing; by changing distribution differently for every token, aren’t we changing the logits necessarily?
- In Figure 4, what is the reciprocal rank exactly? Is a higher value better?
- One thing that is unclear is how token-frequency neurons are regulating confidence specifically? I understand that they are modulating the output distribution towards unigram frequencies, but this affects both the logit ranks as well as the variance, so the effect on "confidence" specifically is unclear. I would appreciate some clarity on this.

### General comments
- It will be useful to state somewhere that your neuron indexing strategy is `<layer>.<neuron>`, which may be unclear to a new reader.

**Limitations:**

Authors have addressed limitations adequately.

---

> ### Author Rebuttal · Authors · 2024-08-07
>
> Thank you for recognizing that our work “focuses on an important problem”, with experiments that are “convincing, and improve upon the findings of the previous works on which this paper is built upon.”
> We appreciate the many insightful questions you pose in your review.
> We address the weaknesses and the reviewer’s primary questions (Q3, Q4, Q5, and Q7) in this rebuttal message. However, due to the numerous questions posed by the reviewer, many of which require detailed responses, and the character limit for the rebuttal, we address three clarification questions (Q1, Q2, and Q6) in a separate comment.
>
> > **Weakness 1:** The paper would be much stronger if some additional discussion revolving around model and vocabulary sizes was included.
>
> Thank you for your input, we agree that further discussion on model and vocabulary sizes would enhance the paper. We address specific questions related to this issue (Q3 and Q4) below and will incorporate the corresponding answers and additional considerations into the final version of the paper.
>
> > **Weakness 2:** The paper mentions a few existing works on confidence regulation and calibration, but does not tie the work done in this paper within the context of this broader field.
>
> Recent work has explored various strategies to calibrate model output, including ensembling model predictions [1, 2], fine-tuning [3, 4], and prompting models to explicitly express their confidence [4, 5, 6, 7]. While our findings do not immediately suggest a strategy for improving models' confidence calibration, they provide valuable insights into the internal mechanisms that LLMs might use for this purpose. In connection with Burns et al. [8] and Azaria and Mitchell [9], our results suggest that it could be possible to estimate models' confidence states (e.g., overconfidence in specific settings) by tracking the activation values of a small set of neurons. This could potentially lead to new methods for dynamically adjusting model behavior based on real-time confidence estimates.
>
> > **Q3:** Can you hypothesize why Gemma 2B has few to no entropy neurons? & **Q4**: Relatedly, what is the effect of model size [...]? What about the vocabulary size?
>
> These are good questions. We believe that the small presence of entropy neurons in Gemma might be due to two characteristics that differentiate it from the other models considered: the large MLP dimensionality and the large vocabulary size.
> In Gemma, the MLP layers have \~32k dimensions, which is roughly 10x larger than in GPT-2 Small and 3x larger than in LLaMA 2 7B. Given the very large number of neurons, the entropy-regulating function might be implemented in a different way within the final MLP layer.
> The vocabulary used for the Gemma models is also significantly larger than for other models (\~256k tokens, 8x larger than LLaMA’s vocabulary). Such a high dimensionality in the unembedding projection might be in contrast with the presence of a null space in the projection matrix.
>
> In general, a larger vocabulary size implies a higher-dimensionality mapping performed by the unembedding matrix. The projection from a low-dimensional space to a higher-dimensional space makes the presence of a null space in the projection matrix more costly. In other words, dedicating the same number of dimensions to a null space results in a greater loss of representational capacity, making the presence of an unembedding null space (and therefore of entropy neurons) less likely.
>
> One non-architectural but training-related factor that we studied, in relation to the emergence of entropy neurons, is the application of dropout. Even though dropout cannot be the sole factor influencing this phenomenon, as we observe entropy neurons in LLaMA, which was trained without dropout, we observed that it has an effect on the size of the unembedding effective null space (the results are reported in Appendix G).
>
> In conclusion, studying the architectural and training factors that determine the emergence of entropy neurons is an interesting direction for future research that warrants further investigation.
>
> > **Q5:** By changing distribution differently for every token, aren’t we changing the logits necessarily? & **Q7:** One thing that is unclear is how token-frequency neurons are regulating confidence specifically?
>
> Token frequency neurons, as opposed to entropy neurons, do affect the output logits directly, and you are correct in noting that they impact logit ranks. The confidence-regulation function of these neurons is achieved by shifting the output distribution closer to or further away from the token frequency distribution. This token frequency distribution is what the model can default to in cases of high uncertainty.
> This phenomenon is supported by the observed anti-correlation between the entropy of the model's output distribution and its KL divergence from the token frequency distribution. Intuitively, the token frequency distribution represents an educated guess for the next-token prediction (e.g., when little or no contextual information is available), thus providing a baseline confidence level for the model. Token frequency neurons regulate the model's confidence in its predictions by adjusting how closely the output distribution aligns with this baseline.
>
> ---
> In conclusion, we would like to thank the reviewer for their input. We will incorporate their comments and the corresponding answers in the final version of the paper. In particular, we will include the details about the token frequency computation (Q1) in Appendix D and reference it in Section 4. We will add the considerations about the effect of model and vocabulary size on the presence of entropy neurons (Q3 & Q4) in Section 3.4, the definition of reciprocal rank (Q6) in Section 5, and we will correct the neuron notation. Thank you again for the thorough feedback.

---

> ### Author Response · Authors · 2024-08-07
>
> We believe the key parts of the review were addressed in the main rebuttal, but we include this comment for completeness.
>
> > **Q1:** How are the token frequencies computed for the second half of the paper?
>
> The empirical token frequency distribution that we compute is the unigram distribution: the distribution of tokens (i.e., entries in the vocabulary) over the whole training corpus. More specifically, the frequency of token $t \in \mathcal{V}$ over corpus $\mathcal{C}$ is computed as  \# of occurrences of $t$ in $\mathcal{C}$ / $|\mathcal{C}|$. This distribution depends on the training data, and ideally, it should be computed for a specific model using the exact training corpus that the model was trained on. We could achieve this for the Pythia models as we had access to the data statistics of The Pile. For GPT-2, since the exact training data is not available, we used a randomly sampled 500M-token subset of the OpenWebText corpus as a proxy for the original distribution (as mentioned in Appendix G). Different training data distributions might lead to different token frequency distributions. However, we expect a model to align its output distribution with the token frequency distribution it was exposed to during training.
>
> > **Q2:** How many entropy neurons usually exist?
>
> The extent to which a neuron is considered entropy-regulating depends on the fraction of its effect that is mediated by the LayerNorm. This quantity varies on a continuous scale, and defining a specific number of entropy neurons would require setting a hard threshold on this measure, which is somewhat arbitrary and difficult to determine precisely. However, in our analyses, we observed that the number of neurons for which the LayerNorm-mediated effect is substantial typically accounts for around 0.1-0.3% of the MLP dimensionality.
>
> > **Q6:** In Figure 4, what is the reciprocal rank exactly? Is a higher value better?
>
> The reciprocal rank is computed as $ \frac{1}{r}$, where $r \in \{1, \dots, |\mathcal{V}| \}$ is the position of the correct next token in the list of all vocabulary tokens sorted by their predicted probability mass in descending order. For example, if the correct next token is ranked 1st in the predicted probability distribution $P_{\text{model}}$ (i.e., the next token prediction is correct), the reciprocal rank is 1. Lower values indicate that the correct token is ranked lower in the predicted probability distribution (i.e., that the prediction is worse).
>
> > It will be useful to state somewhere that your neuron indexing strategy is <layer>.<neuron>, which may be unclear to a new reader.
>
> Thank you for pointing this out. We will add a sentence to clarify the notation.
>
> ---
>
> [1] Wang, X., et al., Self-Consistency Improves Chain of Thought Reasoning in Language Models. ICLR 2023.
> [2] Hou, B.,et al., Decomposing Uncertainty for Large Language Models through Input Clarification Ensembling. ICML 2024.
> [3] Jiang, Z., et al.,. How can we know what language models know?. TACL
> [4] Kadavath, S., et al.,  Language models (mostly) know what they know. arXiv.
> [5] Lin, S., et al., Teaching models to express their uncertainty in words. arXiv 2022.
> [6] Si, C., et al., Prompting GPT-3 To Be Reliable. ICLR 2023.
> [7] Tian, K., et al., Just Ask for Calibration: Strategies for Eliciting Calibrated Confidence Scores from Language Models Fine-Tuned with Human Feedback. EMNLP 2023.
> [8] Burns, C., et al,. Discovering latent knowledge in language models without supervision. ICLR 2023.
> [9] Azaria, A. and Mitchell, T., The internal state of an LLM knows when it's lying. arXiv 2023.

---

> > ### Comment · Reviewer_C4Wm · 2024-08-13
> >
> > Thank you for your response, some of my queries have been answered. I have gone through the other reviewers' comments and will maintain my score.

---

### Author Rebuttal · Authors · 2024-08-07

We would like to thank the reviewers for their thorough feedback. We are glad they found our work well-motivated, clearly presented, and insightful. We address each reviewer’s points in the respective rebuttal sections. Additionally, we attach a PDF file with the additional results referenced in the response to Reviewer Thsu.

---

### Decision · Program_Chairs · 2024-09-25

**Decision:**

Accept (poster)

**Comment:**

The work studies two types of neurons in the last layer of the transformer models that are responsible for calibrating their predictions. The paper makes a small solid contribution to the field of interpretability. The study would be of more value if conducted or extended on real-world tasks. At the current form of the result, it is unclear how generalizable are the findings presented in the paper.